# From Early Diagnoses to New Treatments for Liver, Pancreatic, Gastric, and Colorectal Cancers Using Carbon Nanotubes: New Chances Still Underexplored

**DOI:** 10.3390/ijms26189201

**Published:** 2025-09-20

**Authors:** Silvana Alfei, Caterina Reggio, Guendalina Zuccari

**Affiliations:** 1Department of Pharmacy (DIFAR), University of Genoa, Viale Cembrano, 4, 16148 Genoa, Italy; 2Laboratory of Experimental Therapies in Oncology, IRCCS Istituto Giannina Gaslini, Via G. Gaslini 5, 16147 Genoa, Italy; caterinareggio@gaslini.org

**Keywords:** carbon nanotubes (CNTs), CNT-improved anticancer therapies, CNT-improved cancer diagnosis, hepatopancreatic carcinoma, intestinal cancer

## Abstract

Pancreatic, liver, gastric, colorectal, and rectal cancers (PC, LC, GC, CRC, and RC) are highly lethal tumours, with a 5-year survival rate of 10.5% (PC) and <20% (LC), and of 5%, 12%, and 13% for IV-stage GC, CRC, and RC, respectively. Currently, PC and LC represent the third leading cause of cancer-related death, while GCs and CRCs account for 4.8 million cancer cases and 4.4 million cancer deaths worldwide. Poor prognoses are mainly due to late diagnosis, limited efficacy of available treatments, tumour recurrence, as well as therapy-induced secondary tumorigenesis and drug resistance. In recent decades, these issues have been afforded using nanomaterials (NMs), with promising results. Carbon nanotubes (CNTs) are nonpareil nano systems, which have demonstrated high potential in both cancer diagnosis and treatment, showing to be excellent vehicles for drugs, antibodies, genes, etc. Used alone or in combination with available therapeutic strategies, such as photothermal, photodynamic, drug targeting, gene, immune, and chemotherapies, CNTs have shown notable results in laboratory settings, enhancing the anticancer effects and reducing toxic outcomes of traditional treatments. Anyway, despite PC, LC, and CRC being three of the five tumours (60%) considered the most perfidious and lethal cancers, studies on the use of these innovative NMs to cure them represent only 37% of those regarding the treatment of the most known tumours. Regarding this scenario of a worrying lack of efficient treatments for highly lethal PC, LC, GC, CRC, and RC, this umbrella review was drawn up to promote filling this gap in studies by reporting the still too limited and often obsolete experimentation on the possible use of CNTs for their diagnosis and therapy. To this end, such case studies have been collected in several informative Tables which are functional for readers, and the studies have been discussed. This study wants to sensitize scientists towards more extensive research to find novel safer applications of CNTs against PC, LC, GC, CRC, and RC, both in terms of early diagnoses and efficient treatments. Such efforts should also focus on clarifying the not yet fully unveiled toxicological aspects and regulatory hurdles, both of which persist around CNTs. Research should also be finalized to produce patents rather than journal articles, thus accelerating the translation of CNTs to clinical practice.

## 1. Introduction

Cancer is a disease characterized by unrestrained cell development without contact inhibition, which in its metastatic forms invades other bodily organs causing dramatic impairments of organism functions [1]. As a direct consequence, the health and quality of life of cancer-bearing individuals is totally compromised [2,3]. Cancer is considered the nastiest and most perfidious disorder of recent decades, whose rate of mortality is at worrying levels [4,5]. Cancer may arise from several sources, such as genetic predispositions, environmental and behavioural factors (exposure to radiation and/or carcinogenic chemicals, active and passive smoke, obesity), and microbiological factors such as viral or bacterial infections, as shown in stomach cancer (*Helicobacter pylori*) and cervical cancer [6,7]. Globally, 20 million new cases of cancer were registered in the year 2022, causing 9.7 million deaths [8]. In that year, patients presented mainly lung (12.4%), breast (11.6%), colorectum (9.6%), prostate (7.3%), and stomach (4.9%) cancers in the indicated percentages [9]. Lung cancer was the leading cause of cancer death (1.8 million deaths (18.7%)), followed by colorectal (9.3%), liver (7.8%), female breast (6.9%), and stomach (6.8%) cancers [9]. In 2050, it is estimated that the global number of new cases of cancer will reach 35 million. Among tumours, gastrointestinal cancers (GICs), comprising gastric cancer (GC), colorectal cancer (CRC), and rectal cancer (RC), account currently for one in four cancer cases and one in three cancer deaths globally, with an incidence of 4.8 million cancer cases and 4.4 million cancer deaths [10]. Although GC incidence is decreasing, that of CRC has increased, especially in formerly low-incidence regions. In fact, among GICs, CRCs were the third most common cancer (1.8 million cases) and the second most common cause of cancer-related death (881,000 deaths) for both men and women worldwide in 2018 [11]. The metastatic forms of GICs collectively have a poor prognosis, with a 5-year survival rate of only 5% for IV-stage GC, only 12% for stage IV or metastatic CRC, and of only 13% for rectal cancer (RC) at stage IV [12].

Furthermore, as the twelfth most common cancer and the sixth leading cause of cancer-related death worldwide, pancreatic cancer (PC) is a highly lethal disease [13], with a very low overall survival rate [14]. In a study carried out on 625 PC-bearing patients, of which 569 were followed from 1 to 75 months, the median overall survival was 9.3 months. The overall 1-, 3-, and 5-year survival rates were 37.8%, 15.1%, and 10.5%, respectively [14]. PC accounted for 510,992 new cases and 467,409 deaths in 2022 [9]. In the United States, projections for 2024 estimated approximately 66,440 new cases and 51,750 deaths, placing PC as the third leading cause of cancer-related death, surpassing breast cancer [15]. The number of new cases is expected to reach 998,663 by 2050, establishing a nearly 95.4% increase from 2022 and an annual growth rate of 1.1% [16]. Liver cancer (LC) is the third leading cause of cancer-related deaths worldwide, with a global incidence expected to increase by 55% between 2020 and 2040 and a death incidence higher in men than in women [17,18,19,20,21]. The 5-year overall survival rate in LC is still below 20% due to the limited efficacy of currently available treatment options and highly frequent post-treatment tumour recurrence [22]. The incessant fight against cancers has led to the development and improvement of several cancer treatment options, such as surgery, chemotherapy, radiotherapy, endocrine therapy, immunotherapy, phototherapy, gene therapy, and combination therapy, each one demonstrating increasing efficiency in cancer treatment in terms of decreasing cancer-related morbidity and death [5]. Anyway, the optimum has not yet been reached, as the data of low survival rates reported above confirm. Although being promising therapeutic strategies, the available options to counteract cancers still have several limits which impede their absolute effectiveness [23] and lead to poor prognoses, especially for the most lethal and perfidious existing cancers, including lung cancer, CRC, breast cancer, PC, and LC [24]. Limits include severe treatments-related secondary tumorigenesis, the high incidence of relapses, the common incapability of existing treatments to be effective also against cancer cells which have acquired drug resistance, and their several side effects which significantly lower the quality of life of cancer patients during and after treatments and for all of their remaining life [25]. Additionally, the effectiveness of these treatments strongly varies depending on several factors, such as tumour stage, size, location, the presence or absence of metastasis, the state of health of cancer patient, concomitant diseases, as well as his/her age [26]. Primary and secondary preventions, mainly recognised as diminishing obesity, immunizing populations against hepatitis B virus infection, controlling the use of tobacco and alcohol, and screening individuals for possible cancers to realize early diagnosis, play a key role in reducing the incidence of cancers [9,10]. Such approach can effectively lead to millions of future cancer diagnoses and many lives saved worldwide over the upcoming years [9,10]. Preventive interventions in LC at-risk patients are rational strategies to substantially improve the poor prognosis for the patients [27]. Anyway, both in terms of cancer early diagnosis and treatment, novel, more effective, and less toxic anticancer approaches which function regardless of the emergence of multidrug resistance (MDR) in cancer cells are urgently necessary to ameliorate life conditions of cancer patients and reduce the pain associated with currently available treatments [28]. Unfortunately, this is an era where despite incessant advances scientists are still not completely capable of counteracting a severe, deceitful, and lethal disease such as cancer due to the limited or poorly efficient arsenal at their disposal [5].

### Methodology to Create This Review and Related Tables

Concerning the collection of materials to create this review and construct Tables, a first search was made on both Scopus and PubMed database, crossing first the keyword “hepatopancreatic cancer, liver cancer, pancreatic cancer, gastric cancer, colorectal cancer, intestinal cancer, carbon nanotubes”. Then, we cross similar keywords, substituting cancer with tumour. Results by these two procedures were unified for the searches made on both PubMed and on Scopus, obtaining two collections of articles. The collection accounting for the high number of studies was maintained, which was that which derived from research found on Scopus. From all articles found, review articles and obsolete studies were removed; then, only experimental works and preclinical and clinical trials, if present, were considered. Since to construct graphs available in Section 7 of this paper we used data reported in Tables contained in our previous study [5], we make present to readers that, concerning such Tables, database and criteria for selecting the articles to be used were the same as just described, but keywords were different. Specifically, we cross first the keywords cancer therapy, tumour therapy, cancer diagnosis, tumour diagnosis, cancer biosensing, tumour biosensing, and carbon nanotubes. Then a second search was made using the same database, crossing the keywords photodynamic therapy, photothermal therapy, immunotherapy, gene therapy, chemotherapy, combined therapy, and carbon nanotubes.

## 2. Nanotechnology for Improved Cancer Treatment and Diagnosis

In this context of a limited and poorly efficient arsenal available to treat and diagnose cancer, nanotechnology has significantly contributed to transforming oncology by providing innovative therapeutic alternatives to overcome the limitations of conventional ones [29]. The transformative potential of nanotechnology in cancer diagnosis, treatment, and drug delivery, as well as the use of sustainable nanocomposites derived from natural sources such as plants and microbes, have been reported in relevant reviews [30]. Eco-friendly and biocompatible nanocomposites have demonstrated improved therapeutic efficacy, reduced environmental impact, and have met green chemistry principles [31,32]. Due to their unprecedented skills in administering and delivering drugs, several nanomaterials (NM)-improved anticancer drug delivery systems (DDSs), including solid-lipid nano capsules, polymeric micelle dendrimers, magnetic nanoparticles, gold nanoparticles, and liposomes, have been developed [32,33].

A good number of these NMs, which have remarkably enhanced the efficiency of anticancer chemotherapeutics [1,34], have been already approved for clinical use and are on the market [30,35,36]. Moreover, several cutting-edge cancer treatments, in the form of core/shell nanomaterials and multi-material nanocomposites, such as inorganic/organic, inorganic/inorganic, organic/organic, and organic/inorganic composites [37,38,39,40,41,42,43,44,45,46], have positively transfigured the situation existing before the advent of nanomedicine [47,48,49,50]. Additionally, nanotechnology has allowed the development of nanocomposites for nanomaterial-improved cancer diagnostics and imaging. Over the past decades, quantum dots (QDs), gold nanoparticles (AuNPs), cantilevers, and other NPs have shown great potential for diagnostic applications, and nano diagnostic assays have been developed. A QD-based assay capable of detecting biotinylated prostate-specific antigen (PSA) at 0.38 ng/L and a bio-barcode assay capable of detecting 30 attomol/L PSA in a 10-μL sample have been developed [51]. Anyway, although nano diagnosis promises increased sensitivity, multiplexing capabilities, reduced costs, and intracellular imaging for many diagnostic applications, no clinical use exists so far [51]. Further work is necessary to fully optimize diagnostic nanotechnologies for clinical laboratory settings [46].

CNTs have a nanosized diameter of a few nanometres and a micro sized variable length [52]. As reported in the literature, CNTs are nanomaterials with exceptional electrical, mechanical, optical, and thermal properties, the application of which could remarkably improve performance in several sectors [53]. An extensive and detailed description of the possible structures (SWCNTs, MWCNTs, and DWCNTs) and conformational assets of CNTs, their physicochemical and biomedical nonpareil properties, and methods to prepare them, including the eco-friendly pyrolysis of biomass, is available in our recent papers [5,52,54,55]. Nowadays, several in vitro and in vivo applications of CNTs as anticancer devices and/or adjuvant ingredients in anticancer therapies have been reported at least in laboratory settings [5]. CNTs could serve as excellent and efficient transporters for the delivery of entrapped drugs, including anticancer compounds, biomolecules, and genetic material, thanks to their cylinder-like and fibre-like structures which allow high drug loading capacity and elevated cells penetration, respectively [5,56,57].

Collectively, adequate functionalization has furnished modified CNTs, which in addition to being capable of carrying anticancer drugs and proteins for CNT-improved chemotherapy [58] possess the capacity of carrying and delivering genes for CNT-enhanced anticancer gene therapy or antibodies for enhanced anticancer immunotherapy [5,59].

CNTs have demonstrated great potential against difficult-to-treat cancers such as triple negative breast cancer (TNBC), which is a malignant cancer with a very high mortality rate around the world and very few existing efficient anti-TNBC drugs to treat it [60]. Preliminary in vitro and in vivo studies by Asadipour et al. on using CNTs to manage TNBC showed increased cell death and reduction in spheroid numbers in the CNT-treated cancer cells in comparison to control. Additionally, a significant reduction in the tumour volume in the TNBC-model mice was observed in the CNT-treated animals compared to the untreated animals [60]. Moreover, CNTs loaded with ginsenoside Rg3 (Rg3@CNT), prepared by Luo et al., were revealed to possess a cytotoxic activity based on an apoptotic mechanism which was higher than that of Rg3 alone in vitro. Additionally, Rg3@CNT triggered a higher antitumour immunity, and attenuated the TNBC cell growth in vivo. On these considerations, Rg3@CNT may be applied as a new potential therapeutic strategy for immunotherapy against TNBC [61].

Aiming at achieving a direct extermination of cancer cells, CNTs have also been utilized as enhanced photosensitizers for CNT-improved photothermal therapy (PTT) and photodynamic therapy (PDT) [5,59,62,63]. High concern persists regarding the scarce solubility of CNTs in most solvents, which limits their dispersity in water and their use in manufacturing CNT-based nanocomposites [64].

Moreover, despite CNTs possessing extraordinary physicochemical, mechanical, electrochemical, and electrochemiluminescence properties different from those of nanomaterials known so far [65], their possible nanotoxicology and their regulation are still not clear [5,54,55].

To address solubility and toxicological issues, several surface modifications have been carried out to increase their hydrophilicity and water dispersibility and decrease toxicity [64,66,67]. In addition, advanced synthetic methods and highly efficient purification procedures, which provide more perfect CNTs without structural defects and impurities, have already been developed to limit CNTs’ toxic effects [52]. Furthermore, preventive behavioural conduct, especially in working settings, is continuously advertised to prevent unconscious and hazardous improper use [52]. Anyway, an increased number of studies should be implemented to make their large-scale production safer and allow for no-risk extensive utilization, especially in biomedicine. Figure 1 summarizes the main advantages and disadvantages of utilizing CNTs in biomedical applications [1].

In collecting information for editing our latest study concerning the use of CNTs in the treatment and diagnosis of the most common types of cancers [5], it was evident that despite PC, LC, and CRC are reported as the among the five tumours (60%) considered the most perfidious and lethal cancers affecting humans worldwide studies on the use of CNTs to treat them (40) represent only a minimal part of those regarding the CNT-based treatments of the most prevalent tumours (107).

In this regard, due to the worrying lack of efficient treatments and diagnosis methods for these highly lethal and aggressive cancers and for the early detection of GC, the poor prognosis of which is mainly due to late diagnosis, this umbrella review was edited. Precisely, the still too limited and often obsolete experimentation on CNT applications for the diagnosis and therapy of PC, LC, GC, and CRC were discussed. To this end, the existing case studies on this topic found in the literature have been provided and discussed by using several informative and easy understandable Tables. Differently from the already existing reviews on the use of CNTs for treating cancers in general, the major scope of this study was to inform scientists about the high positive impact that CNTs could have if applied in the therapy and diagnosis of specific tumours, such as PC, LC, GC, CRC, and RC, in relation to the few and obsolete articles existing on the topic so far. Secondly, this paper aimed at sensitizing researchers towards more extensive research to find novel possible applications of CNTs against these difficult-to-treat malignancies, both in terms of early diagnosis and efficient treatment. In our opinion, such research should be focused more on the development of patents than to that of journal articles. We are strongly hopeful that the information provided here and the possible future patents inspired by this work could accelerate the translation of CNTs into clinical practice, which is still hampered by the not yet fully unveiled toxicological aspects, regulatory hurdles, and other factors, which have also been reviewed here.

## 3. Carbon Nanotubes for Anticancer Therapy and Cancer Diagnosis

The following Figure 2 shows a roadmap for the evolution in the use of carbon nanotubes in cancer targeting and diagnosis over the years 2004–2022, as reported by Singh and Kumar [1].

In anticancer biomedicine, CNTs have been widely studied as vectors for the transport and delivery of various agents, including contrast media, antigens, nucleotides, plasmids, therapeutic agents, etc. [5]. In these studies, drug-based, nucleotide-based, plasmid-based, and antigen-based CNT complexes and/or nanocomposites have been developed for enhanced anticancer chemotherapy, gene therapy, and immunotherapy [5]. Moreover, their synergistic effects in combination with other cytotoxic agents have also been studied [1]. Collectively, in anticancer drug delivery CNTs have been demonstrated as superior to other carbon nanomaterials (CNMs), such as fullerenes, mainly due to their chirality and diameter-based physicochemical properties which confer CNTs high stability and drug loading capacity (DLC), and to their needle-like shape which provides CNTs an enhanced cell penetration capacity [1,5]. In this regard, Joshi et al. studied the delivery of methotrexate (MTX) in vitro to MDA-MB-231 breast cancer cells, using as carriers both aminated multi-walled carbon nanotubes (MWCNTs) and aminated fullerenes and the results observed results were compared [69]. Although in vitro experiments showed that MWCNTs-MTX elicits lower cytotoxic effects and shorter circulation times than fullerenes–MTX, a significantly better bioavailability for MWCNT-MTX than for MTX–fullerenes was forecast. In fact, when experiments were carried out in vivo, MWCNT-MTX displayed higher anticancer effects at lower doses and minor haemolytic toxicity than MTX–fullerenes [69]. Moreover, CNT-based nanocomposites have been extensively used in experiments as enhanced photosensitizers, in conjunction with various other anticancer techniques, such as PTT, PDT, and sonodynamic therapy (SDT), for combination treatments of various types of cancer [1].

In fact, in addition to surgery and traditional chemotherapy (CT), other current available therapeutic strategies include anticancer PTT, PDT, gene therapy (GT), immunotherapy (IT), and their association to realize combination therapy (COT) with enhanced outcomes due to synergistic effects [5]. Unfortunately, conventional light- and photosensitizers necessary to perform PDT and PTT are affected by several drawbacks, including undesired side effects on skin, scarce targeting to cancer cells, and poor efficiency in hypoxic tumour microenvironment TME. In this context, CNTs may represent outstanding next-generation light- and photosensitizer agents for more effective PDT and PTT, due to their unprecedented better photophysical properties, broad electromagnetic absorbance spectrum [70], and their ability to head specifically to cancer cells and amass in the tumour site. Although traditional photothermal agents and photosensitizers (PSs) are intrinsically endowed with high photothermal conversion efficiency and optimal stability for biological applications, the unique structure of CNTs allows one to load them with high efficiency by π–π stacking noncovalent functionalization, thus providing CNT-based PSs with higher biocompatibility, bioavailability, and thermal effects than pristine PSs while reducing undesired outcomes [70]. Moreover, CNT-based tumour-targeting nanocomposites can be used in combination with PTT treatment, thus leading to more accurate and efficient tumour elimination [5]. Gene therapy (GT) is another therapeutic strategy to counteract cancer. It is based on the replacement and/or silencing of specific genes, as well as on the administration of oncolytic genes to cancer-affected patients using proper carrier. GT includes also CAR-T cell therapy and CRISPR-Cas9 gene editing [71]. When not viral, the selected carrier should possess precise requisites for making gene therapy efficient, primarily the capability to form stable complexes with genetic material [71,72,73]. Once the complex genes@nanocarriers is formed, it is administered to patients and is internalized in cancer cells by endocytosis. Once inside the cancer cell the complex remains trapped inside an endosome from which it must rapidly escape. The not fast endosomal escape of the genes@nanocarriers complex, due to an insufficient buffer capacity of the nonviral nano vectors, can translate into the lysosomal early degradation of the nanocomplex and poor transfection in vivo, thus invalidating the anticancer therapeutic effects hoped for [74]. CNT-based vectors, opportunely modified to improve their buffer capacity, can advance endosomal fast escape and survival of the gene complex, thus significantly upgrading transfection efficiency. Moreover, CNTs could be best platforms for performing effective IT. Polymer-modified CNTs have been demonstrated to be excellent immune adjuvants which can promote dendritic cells (DCs) maturation and CD8^+^ T-cells infiltration in TME, where they can also release antitumour factors [75]. Furthermore, peptide-conjugated MWCNTs can trigger cytokine secretion and stimulate T-cell differentiation and proliferation [76]. Collectively, CNTs can reinforce antitumor immunity in cancer-bearing patients via multiple mechanisms of modulation of the immune system [5]. Several examples of case studies on the application of CNTs for improved PTT, PDT, GT, IT, CT, and COT against the most diagnosed cancers have been recently reported by collecting them in several informative and serviceable Tables and discussing them [5]. Therefore, thinking it redundant to report them here again and due to them not being in the scope of this new study which is focused specifically on hepatopancreatic and intestinal tumours, we have preferred to provide Table 1, in which the main reported cytotoxic effects of CNTs are compared with those of several other nanomaterials. Table 1 contains results from both in vitro and in vivo studies, carried out on different cell lines and animal models [77]. Specifically, three types of CNT-based materials, reported in rows 8–10, have been compared with several types of inorganic nanoparticles commonly used in experiments for biomedical applications and fullerenes. Table 1 reports data which shows that the possible toxicity of nanomaterials, including CNTs, strongly depends on the type of cell and/or animal model, time of exposure, concentration, administration route, nanoparticles dimensions, and even the type of assay. Anyway, it is almost impossible to find a linear correlation between at least two of these variables. Additionally, a direct comparison between the data of cytotoxicity in Table 1 for the different nanomaterials is difficult since all the studies were carried out separately in different conditions and parameters. The discussion of the limitations (toxicity, clinical lag, reproducibility) of CNTs, which affects the number of ongoing clinical trials and hampers and delays their effective translation to clinical settings, was expanded later.

Almost all studies on the use of CNTs to manage cancers established that they can be used to develop multimodal platforms capable of efficiently realizing the simultaneous early cancer detection and enhanced cancer treatment [1]. Notably, to ensure optimal therapeutic outcomes it is of paramount importance to detect cancer in its early stages. Gastric adenocarcinoma (GAC), although not considered among the five most lethal tumours, carries a very poor prognosis, mainly due to the late stage of diagnosis [78]. Early diagnosis of cancer is linked to improved patient conditions, low-dose less-aggressive therapies, and increased life expectancy. In this context, biosensors may serve as accurate diagnostic instruments for early cancer detection [79,80]. Particularly, due to their nonpareil properties, including fast electron transfer kinetics, a wide electrochemical stability window, and large length-to-diameter aspect ratios, SWCNTs are widely experimented on in biosensing applications and have demonstrated great potential as nanoscale building blocks for nano sensors fabrication [81]. As examples, Fan et al. engineered a portable, low cost, high accuracy, and small size MWCNTs smartphone-based electrochemical sensor for revealing CA125, which is a crucial tumour marker frequently found in ovarian, breast, and lung cancer [82]. By using a differential pulse voltammetry (DPV) measurement scheme, this detection system could replace cumbersome and costly hospital instruments, simplify the measurement steps, and shorten times for detection. Furthermore, Lv et al. immobilized bimetallic rhodium@palladium core–shell nano dendrites (Rh@PdNDs) over sulphate-activated MWCNTs, achieving a nanocomposite (Rh@Pd NDs@MWCNTs) which was used as an electrochemical immunosensor. It detected CEA, an important biomarker for tumours, in the low concentration range of 25 fg/mL–100 ng/mL [83]. Also, CA19-9 was detected by a MWCNT-based electrochemical immunosensor (chitosan@CA19-9-antibody@MWCNT@Fe_3_O_4_), with a limit of detection (LOD) of 0.163 pg/mL [84]. Readers particularly interested to know more about the most adopted CNT-improved sensing techniques experimented on in recent years for the early diagnosis of various cancers and their treatment can find such information in a recent paper [5]. Like for the CNT-based anticancer treatments, the related main relevant studies carried out in recent decades on the use of CNT-based sensing systems in the early detection of several tumours through different sensing techniques have been presented in the paper using several informative and easily understood Tables [5]. Additionally, a Table (Appendix A) summarizing studies on the detection of biomolecular markers of other cancers in addition to those which are the topic of this paper by CNT-based biosensors has been reported in Appendix A.

## 4. Hepatopancreatic Cancer Therapy and Diagnosis Using CNTs

### 4.1. Carbon Nanotubes for Pancreatic and Hepatocellular Cancers Diagnosis

Early detection is essential for reducing mortality from pancreatic and hepatocellular cancers, but conventional screening methods can be invasive, expensive, time consuming, and have low efficiency [85]. Non-invasive diagnostic approaches hold promise for more efficient cancer surveillance, including liquid biopsy technologies targeting KRAS mutations, exosome markers, and VOC breath analysis for efficient pancreatic cancer (PC) detection. Cancer circulating ctDNA methylation panels plus AI-driven radiological assessments allowed evolutions in hepatocellular carcinoma (HCC) surveillance [85]. These innovations address long-standing challenges in early PC and HCC diagnosis by increasing sensitivity and patient comfort [85]. Anyway, to meet the need of novel methods for the early diagnosis of such tumours, the outcomes of which could depend less on the clinicians’ ability, new CNT-based nanomaterials have been developed for theragnostic purposes due to their lightness, high surface area, considerable drug loading capacity [86], and high possibility of functionalization with biological macromolecules without structural deterioration but with enhanced outcomes.

#### 4.1.1. Carbon Nanotubes for Pancreatic Cancer Diagnosis

As above reported, PC is an aggressive lethal tumour, with a low survival rate which reaches only 5–7% in its IV-stage advanced forms. Since PC is refractory to conventional chemotherapy regimens it is often treated with surgical resection [87,88], the success of which is hampered by PC rapid grow and fast invasion of the adjacent tissues, thus inducing metastasis to other organs at a distance [89,90]. Worryingly, since clinical signs are missing at the initial phases of the disease most PC patients are diagnosed only at the advanced stages of cancer when metastases are already present, thus translating into an even worse prognosis. Endoscopic ultrasonography (US) and computational tomography (CT), associated with fine-needle biopsy, have been extensively applied for the detection and staging of PC [91,92]. However, especially in CT, diagnosis outcomes are highly dependent on the clinicians’ ability. Therefore, the need to find novel methods for the early diagnosis of PC is extremely urgent and so new CNT-based nanomaterials have been developed for PC theragnostic for the reasons already exposed in the previous section [86]. Several nano–bio hybrids have been developed and tested in laboratory settings, enabling the real-time detection of specific biomarkers of PC with high sensitivity [93]. In this context, several studies have been carried out for the early diagnosis of various cancers through the detection of cancer biomarkers such as carbohydrate antigen 19-9 (CA19-9), which is the main marker of pancreatic, bile duct, liver, stomach, and colorectal cancers. Jin et al. deposited MWCNTs on a microporous filter paper, thus engineering a MWCNT-based biosensor for CA19-9 detection which exhibited a wide detection range (0–1000 U/mL), with linearity maintained also at the minimal concentration of CA19-9 and 30-fold higher detection speeds [94]. A simple, low cost, fast, and green microwave-assisted synthetic method was used by Alafarj et al. to prepare a glucose-derived carbon quantum dots/gold (CQDs@Au) fluorescence nanocomposite. Upon affixion of the anti-CA 19-9-labelled horseradish peroxidase enzyme (Ab–HRP) on its surface, subsequently trapped by another monoclonal antibody, it was used as an immune-sensing device for the detection of CA 19-9 in human serum [95]. The fluorescence intensities of the successful immunoassay sensing device were proportional to the CA 19-9 antigen concentration and were linear in the range of 0.01–350 U/mL, with a LOD of 0.007 U/mL [95]. This device could be potentially used to screen patients with an increased probability of acquiring PC. Anyway, since diabetic patients can also exhibit elevated levels of CA19-9, thus causing false positive results, a cut-off of 75 U/mL was determined by Murakami et al. to separate PC patients from diabetic ones [96]. Also, Thapa et al. developed a SWCNT-based immune sensor to detect CA19-9 through impedance spectroscopy by anchoring CA19-9 antibodies on the surface of CNTs [97]. CA19-9 was selectively detected at the low concentration of 0.35 U/mL, which might increase the survival rate in PC-affected patients [97].

PC can also be diagnosed based on the amount of guanine and deoxy guanine triphosphate (dGTP) residues present in the DNA of PC patients recruited. To this end, Liu et al. developed MWCNT-modified glassy carbon electrochemical sensors, which were successful in PC diagnosis when combined with the technique of random amplified polymorphic DNA (RAPD) [98]. Since insulin-like growth factor receptor R1 (IGF-R1) is overexpressed in aggressive PC cells, Lu and colleagues used IGF-R1-modified SWCNTs (IGF-R1@SWCNTs) and conjugated cyanine 7 (CY7) to the obtained nanocomposite to provide fluorescence properties, achieving the final IGF-R1@CY7@SWCNTs. The authors tested it in enhanced PTT treatment against PC by optical guiding imaging [99]. They observed precise PC treatment with minimal toxicity to normal cells, mainly due to the fluorescence properties of IGF-R1@CY7@SWCNTs, which enabled the proper positioning of the laser and the monitoring of such an intervention procedure [99].

#### 4.1.2. Carbon Nanotubes for Liver Cancer Diagnosis

Human liver cancer is the fifth most common type of cancer globally, with more cases in Asia and Africa than in Europe [100]. Hepatocellular carcinomas (HCCs) or malignant hepatomas are the most common types of liver cancer, accounting for approximately 75% of liver cancer cases [100]. Surgery and liver transplantation are the most used conventional treatments for HCCs. However, although effective at early stages of diseases they fail in advanced HCC cases, leaving very low life expectancies for LC patients [100]. Alternative treatments involve chemotherapy, but treatment-induced toxicity and secondary tumorigenesis, the emergence of multi-drug resistant (MDR) cells, and poor kinetics of conventional chemotherapeutics limit the effectiveness of this approach. Moreover, since for the most part, chemotherapy agents were utilized in HCC anti-angiogenesis drugs, the function of normal cells such as bone marrow, gastrointestinal cells, and hair follicles in the body can be detrimentally impaired [100]. In this scenario, the development of new diagnostic and therapeutic devices is urgently needed for the early detection of HCCs before low survival rate stages occur. In this context, nanotechnology has given its precious and nonpareil contribution, allowing the development of new diagnostic and therapeutic nanocarrier-based systems. Among available NMs, in addition to having furnished efficient devices to treat HCCs by different therapeutic strategies, CNTs have been applied to fabricate novel diagnostic tools for liver cancer. Although in the case of liver cancer the research on CNTs is mostly focused on the development of treatment strategies, and studies regarding liver cancer diagnosis have gained lower attention, various CNT-based methods for LC detection have been anyway experimented.

Li et al. developed a novel immune-sensing method for simultaneously detecting three LC biomarkers, such as α-fetoprotein (AFP), α-fetoprotein variants (AFP-L3), and abnormal prothrombin (APT), by immobilizing proper antibodies on gold-coated CNTs (Au@CNTs) and labelling the obtained nanocomposite with redox probes (anti-Au@CNTs@rdx) [101]. Years later, similarly to Li et al., Hu and co-workers by bio-orthogonal reactions conjugated CNTs with the modified and self-assembled horseradish peroxidase (HRP) and alkaline phosphatase (ALP) for detecting simultaneously the Golgi protein 73 and α-fetoprotein as hepatocellular carcinoma biomarkers (Figure 3) [102].

Due to the use of bio-orthogonal reactions, the biosensor possessed regulable LOD for Golgi protein 73 and α-fetoprotein, while the amount of HRP and ALP on the sensor affected its sensitivity. LODs were in the unprecedented order of tens of pg/mL toward HRP and ALP [102]. Also, the attachment of biomarkers to anti-Au@CNTs@rdx produced noticeably amplified signals, allowing for the diagnosis of LC using several immune-sensing probes. By the capacity to detect a panel of LC-typical biomarkers, non-specificity was avoided. The method had a low LOD and wide linear ranges, thereby enabling quantification of the disease state [101]. A patterned CNT-based surface coated with collagen was engineered by Kucukayan-Dogu et al. and used to identify the invasive properties of LC cells, with respect to normal ones, by evaluating the capability of well-differentiated cells (HUH7) and poorly differentiated cells (SNU182) for attaching the developed patterned CNT surface [104]. Phenotypic differences between cells allowed normal cells to attach better to CNT surfaces than LC ones, permitting for the diagnosis of the invasion level of the LC cells [104]. Tavakkoli et al. merged MWCNTs with maize protein zein and used the obtained nanoparticles (NPs) to engineer a novel nano-bio-composite (maize@MWCNT) to detect and monitor H_2_O_2_ in HepG2 LC cells as a reliable biomarker for LC [105]. Zein enhanced the electron transfer capacity of MWCNTs, thus improving their conductivity and catalytic activity. Using this nano sensor, H_2_O_2_ was successfully detected in a synthetic urine sample and its release from human dermal fibroblasts and human HepG2 cells was monitored [105]. This solution could distinguish the cancer cells (HepG2) from normal cells (HDF), making it a promising sensor for cancer cell diagnostics.

The various methods of sensing PC, HCC, and LC using CNTs are summarized in Table 2 and Table 3. Table 2 reports also data concerning the range of linearity and LOD.

#### 4.1.3. Discussion on the Case Studies Previously Reported

Collectively, at least in laboratory settings, the use of CNT surfaces to anchor antibodies or other elements to detect CA19-9 biomarkers at very low concentrations could be an appealing strategy to realize the early diagnosis of PC and to distinguish between PC patients and those with other diseases in which CA19-9 is present. Chemical interactions have been reported as the most responsible factors for CA19-9 adsorption. They are tuneable by modifying surface physicochemical properties of CNTs [97]. In this context, further studies focused on the manipulation of the molecular recognition elements on the CNT-based sensors allowed the development of novel CNT-based biosensors with higher specificity and sensitivity for early detections and treatment of PC. Such robust biosensors could revolutionize clinical diagnosis of PC, avoiding the need for blood samples while instead making use of bodily fluids such as saliva. It is noteworthy that the salivation detection method is non-invasive and could reduce the cost and time involved in cancer diagnosis. Explorative experiments in this sense could be designed by applying the experimental procedure outlined above, with CA19-9 biomarkers diluted in artificial saliva to test the effectiveness of the CNT-based sensors in their detection. Studies evidenced that CNT-based nanocomposites could also be a new method for studying the differential genes expressed in complex diseases other than cancer, thus providing a valid example of applying CNTs to clinical gene study. The study of Lu et al. confirmed that SWCNTs can function well both as CNT-improved biosensors for tracing PC and as PTT enhancers allowing a PTT tumour-precise targeted therapy, thus paving the way for superior imaging and treatment of PC going forward. Moreover, CNTs have showed that if opportunely functionalized they can allow the detection of multiple LC biomarkers simultaneously, thus allowing the decreasing of LOD values for each biomarker and avoiding false positive results if the concentration of a single biomarker is too low. The study from Kucukayan-Dogu et al. evidenced that physical topography of the surfaces of CNTs, inserted as ingredients in biosensors for LC detection, plays an important role for HCC attachment. The metastatic grade of cancer cells, which could be detected in a fast and facile way using CNT-based sensors developed by the authors, is of paramount importance to determining the most effective therapy strategy depending on the tumour characteristics. From case studies reported in both Table 2 and Table 3, it is evident that when not considering not structurally specified CNTs (7 out of 16 cases) the most applied CNTs for PC and LC diagnosis were MWCNTs (8 vs. 1 case), thus highlighting that MWCNTs were preferentially chosen by researchers for their experiments in biosensing. This is because the use of MWCNTs in sensor development has its advantages, including better mechanical strength and thermal stability than SWCNTs, which is important during sensor development. Furthermore, MWCNTs are cheaper to be produced in bulk synthesis without the use of any catalyst material and can be integrated with a suitable polymer or composite to further enhance the mechanical, physical, and chemical properties. Such integration is more difficult with SWCNTs due to their low solubility. Moreover, SWCNTs generally produced in the presence of catalysts have poor purity, thus are more prone to develop toxic effects and are more expensive. Also, it was also observed that MWCNTs show better performance in rogue and corrosive environments, thus being more suitable for experimentation in complex and real-world samples, without any effect on sensor sensitivity and performance. SWCNT-derived biosensors are known to have limited specific surface area to interact with larger bioelements like mammalian cells, have an uncontrolled manufacturing process, and undergo chemical modification.

Anyway, due to their amazing electronic properties, SWCNTs have gained great attention in electrochemical biosensing.

### 4.2. Pancreatic Cancer Therapy

Pancreatic cancer (PC) is one of the most lethal cancers, the poor prognosis and low-rate survival of which mainly depend on the emergence of chemoresistance and frequent relapses [112]. Before the advent of targeted drug delivery systems based on nanoparticles, as well as thermal, light-based, and acoustic ablation therapies such as PTT, PDT and PAT, the primary clinical treatment involved the surgical excision of the tumour. Unfortunately, as most of the patients were affected by advanced stage cancer, surgeries had full success only in 20% cases [112]. Additionally, in PC there is elevated presence of highly fibrotic stroma (FS), caused by a pancreatic satellite cell (PSC)-induced abnormal deposition of a dense complex of extracellular matrix proteins near tumour cells which can constitute most of the PC mass [113]. FS represents a key factor of patient’s low survival [114,115] because it acts as a physical barrier against drug delivery by distorting the tumour-related blood vessels [116,117]. Moreover, FS can induce resistance to chemotherapy by triggering hypoxia in PC [117]. Anyway, since they are involved in the production of this unusual dense fibrosis [118,119], they could be another important cellular target to inhibit fibrosis and tumour-inducing signalling pathway in PC [120]. An RNA interference (RNAi)-based gene silencing strategy represents an advanced promising therapeutic opportunity to treat PC. RNAi molecules such as small interfering RNA (siRNA) can target and block the expression of distinct genes that are not blocked by other therapeutic strategies such as immune therapy and chemotherapy. To have a better opportunity to overcome the acquired chemoresistance of PC cells, reduce PC relapses, limit the emergence of secondary tumorigenesis induced by severe and protracted chemotherapy, and alleviate PC-bearing patients’ pain, researchers started to study the impact that nanomaterials (NMs) could have in the treatment of PC. Since CNTs have increasingly gained huge interest in the field of both cancer diagnosis and treatment due to their nonpareil properties, they were assessed alone or in combination with available therapeutic strategies against PC, demonstrating excellent results in laboratory settings. Moreover, CNTs have shown the capability to deliver drugs into the dense and fibrotic abnormal condition existing in PC, thus further minimizing the issue of chemotherapy resistance. In the field of gene therapy, viral vectors, usually adeno-associated viruses and retroviruses, possess a sufficiently high transfection efficiency which could ensure a successful outcome, but also present main drawbacks including immunogenicity and paradoxically carcinogenicity [121]. Conversely, nonviral vectors can limit preparation costs and undesirable side effects, while they can also allow the simpler delivery of protein/nucleic acid complexes such as Cas9. Nonviral vectors can be categorized into organic materials, including liposomes, polyethyleneimine (PEI) and derivatives, cationic polypeptides, dendrimers and derivatives, chitosan and derivatives, polyurethane, cyclodextrin and derivatives, etc. [121]. Conversely, inorganic materials encompass gold nanoparticles (AuNPs), carbon-based nanostructures, upconverted nanoparticles, and silica nanoparticles [121]. In general, carbon-based nanoparticles such as CNTs, due to their biocompatibility, large ratios of surface to volume, ease of functionalization, and suitable optical features, have much to offer for gene therapy [121] compared to other cited nonviral materials. Carbon nanostructures, properly surface-functionalized, can deliver various DNA fragments, including large DNA molecules (plasmid DNA (p-DNA)), ribonucleic acid (RNA) molecules (small interfering RNA (siRNA), short hairpin RNA (shRNA), micro RNA (miRNA), antisense mRNA, and small DNA molecules (iC9 suicide gene, antisense oligonucleotides) to the specific target, outperforming both viral carriers and other nonviral nanomaterials [121]. However, several challenges regarding the release and delivery of siRNAs into cells remain still unsolved. Furthermore, although nanotechnology-based techniques have offered promising ways to deliver siRNA to tumour cells, and different preclinical studies have investigated the application of non-viral NPs to deliver siRNAs to PC cells [122], clinical trials are required to assess NMs and CNTs clinical potentials and safety profiles. Table 4 collects relevant studies in this context.

Iancu et al. non-covalently functionalized MWCNTs with human serum albumin (HSA) and assessed the efficiency of the achieved HSA@MWCNT complex in the photothermal ablation of PC cells by administering it intra-aerially to the resected cancer tissue (ex vivo) [123]. The authors observed a substantial and selective accumulation of HSA in the PC cells and under laser radiation: PC cells in the tissues died while the surrounding normal cells remained unharmed [123]. The same year, Karmakar et al. covalently coupled SWCNTs to the epidermal growth factor (EGF) and examined its SWCNT-mediated delivery to PC cells in vitro. A significant uptake of the EGF-SWCNT only 30 min after administration was observed [124].

Later, Mahmood et al. co-administered SWCNTs and etoposide, a drug used commonly in chemotherapy to treat several types of tumours, observing a synergic effect [125] which could allow for the use of a lower dosage of etoposide, thus limiting its well-known toxic effects and the emergence of chemoresistance triggered by a protracted use of high dosage [133,134,135]. Also, Andreoli et al., upon optimization experiments to choose the more efficient molecular weight of polyethyleneimine (PEI), engineered by covalent linkage PEI 600 and 1880 Da-SWCNTs conjugates (PEI600@SWCNTs and PEI1800@SWCNTs) and tested the cytotoxic effects both in vivo and in vitro. In vitro experiments showed that PEI@SWNTs conjugates were able to pass through the biological barriers of cytoplasmic and nuclear membranes, demonstrating in principle to be excellent vectors to deliver therapeutics in PC cells. Also, intense necrosis of the PC cells was observed in vivo in BxPC3 cells when PEI-SWCNT was administered in mice [126].

Conversely, polyethylene glycol (PEG) was used by Mocan et al. to modify MWCNTs, obtaining PEG@MWCNTs nanocomposites which were used as photosensitizers to induce hyperthermia under laser radiation and realize a CNT-improved PTT anticancer treatment against PC cells [127]. Depolarization of the mitochondrial membrane and accumulation of reactive oxygen species (ROS) causing apoptotic death of PC cells were revealed, suggesting the potential of this novel therapeutic strategy to overcome the chemotherapeutic resistance of cancer cells [127]. Gambogic acid (GA) was delivered to both breast cancer and PC cells by Saeed et al., using both graphene and SWCNTs as drug delivery systems (DDSs) for GA (graphene@GA and SWCNTs@GA). Through several types of investigation, both CNMs improved the anticancer effects of GA in both cancer lines [128]. Conversely, Anderson et al. exploited CNTs to realize a CNT-improved anticancer gene therapy, entrapping siRNAs into SWCNTs which acted as a DDS to target PC cells [129]. In vitro experiments evidenced improved siRNA transfection and efficient gene therapy. As a result, a mutant gene (K-Ras) in PC cells was downregulated,

### 4.3. Liver Cancer Therapy

Like PC, liver cancer (LC) is another type of very aggressive and lethal cancer which is difficult to treat because of the tumour location deep in the tissue [112]. The prognosis is low and the 5-year overall survival rate in LC is still below 20% due to the limited efficacy of currently available treatment options and highly frequent post-treatment tumour recurrence [22]. Resection followed by chemotherapy schemes could allow a 5-year survival rate in 50% of the cases, but the usual emergence of drug resistance and severe side effects, including grave pain for LC-bearing patients, strongly lower the overall effectiveness of these treatments [112]. In this regard, researchers thought that including CNTs in the available LC treatments and diagnostic methods could have been helpful in addressing at least some of their drawbacks. With the aim at improving the anticancer effects of gene therapy against LC cells, Pan et al. synthetized an FTIC-labelled CNT-based nanocomposite, conjugating the antisense c-myc oligonucleotides (asODN) and poly-ethyleneimine (PAMAM-NH_2_) of different generations to MWCNTs, achieving as-ODNs@PAMAM@*f*-MWCNTs composites which were administered to liver cancer cell line HepG2 cells [136]. Composites entered tumour cells within 15 min and down-regulated the expression of the c-myc gene and C-Myc protein, thus confirming enhanced transfection, and inhibited the cancer cells growth in time- and dose-dependent modes [136]. Collectively, these nanocomposites demonstrated high potentials for applications in anticancer gene or drug delivery and molecular imaging [136]. Iancu et al. studied the concentration- and time-dependent effects of human serum albumin (HAS)@MWCNTs nanoconjugates when applied to PTT to induce apoptosis in HepG2 human hepatocellular liver carcinoma cells. In parallel, authors investigated its effects on normal hepatocyte cells [137]. The results evidenced that the post-irradiation apoptosis was substantially higher in HepG2 cells than in the normal hepatocyte ones (88.24–92.34% vs. 64.3–70.78%) [137].

Later, Ji et al. used chitosan (CHI)- and folic acid (FA)-modified SWCNTs, producing a nanocomposite (CHI@FA@SWCNTs) for the targeted drug delivery of doxorubicin (DOX) in vitro and in vivo [138]. Results indicated that, when delivered by CHI@FA@SWCNTs, DOX exerted higher efficiency and enhanced target properties due to the presence of FA on the nanocomposite’s surface [138]. In vivo examinations demonstrated insignificant cytotoxic effects of SWCNTs on normal tissues due to the presence of CHI, which enhanced their water dispersibility and biocompatibility [138]. Subsequently, with the aim of overcoming the drug- and radio-resistance of hepatocellular carcinoma, Wang et al. loaded the anticancer drug ruthenium polypyridyl complex (RuPOP) onto MWCNTs, achieving the complex RuPOP@MWCNTs which was then administered to LC cells in vitro [139]. An increased cellular uptake and an enhanced anticancer efficiency were observed, and they were associated with improved effects of X-rays which caused apoptosis in cancer cells [139]. Apoptosis was correlated with substantial ROS overproduction which activated the necessary signalling pathways [139]. The next year, Wen et al. utilized in vivo CoMoCAT^®^-SWCNTs@PL@PEG-NH_2_ nanocomposites to trigger strong thermoacoustic shocks in targeted LC cells in mice models under ultra-short pulse microwave radiation [140], thus destroying mitochondria and causing apoptotic death in targeted cancer cells. Experiments showed excellent inhibition of tumour growth with no side effects, thus paving the way for the development of this strategy for treating deep-sealed tumours. Table 5 collects relevant case studies on the application of CNTs in the treatment of LC.

In the past, Gu et al. engineered doxorubicin (DOX) and hydrazine-benzoic acid (HBA) bearing SWCNTs via hydrazine bonding, producing nanocomposites (DOX@SWCNTs and DOX@HBA@SWCNTs) which were tested on HepG2 LC cells to assess their release profiles and cytotoxic effects [141]. Results demonstrated a pH-dependent drug release rate, with maximum releases at the pH = 5.5 of the tumour microenvironments (TME), reaching 50% of the drug released from DOX@SWCNT composite and 73% from DOX@HBA@SWCNTs after 60 h incubation [141]. Higher cytotoxic effects were observed for the DOX@HBA@SWCNTs nanocomposite with respect to DOX@SWCNTs due to enhanced cellular internalization [141].

Sobhani et al. synthesized 80% *w*/*w* PEGylated CNTs (PEG@CNTs) using previously oxidized MWCNTs with sulphuric acid solution (1/3 *v*/*v*), obtaining a PEG@o-MWCNTs nanocomposite which was experimented on using CNT-based PTT to eradicate solid tumours, such as melanoma [130]. Early experiments were performed using both HeLa and HepG2 cancer cells to assess the cytotoxic effects of both MWCNTs, o-MWCNTs, and PEG@o-MWCNTs, observing a progressive reduction in cells death, and thus leading to the researchers forecasting for a low toxicity of the prepared nanocomposite when used alone. Conversely, when PEG@o-MWCNTs was applied in PTT to eradicate melanoma in mice, it caused a notable reduction in melanoma tumour size [130].

As it was anticipated by our survey which inspired this study, the studies available on the experimentations of CNT-based nanocomposites for the treatment of PC and LC are very limited (10 concerning PC and 7 concerning LC), and date back more than 10 years ago. Curiously, although the urgent need for novel treatment options to improve the poor diagnosis of PC and LC and the low survival rates of patients bearing these lethal tumours, and despite their excellent outcomes, these early studies scarcely stimulated scientists in the field to make further in vitro and in vivo experiments to deepen our knowledge about this promising nanotechnology. As confirmation, the subsequent Table 6 collects the limited number of other more recent studies found on the application of CNT-based nanocomposites to deliver drugs or gene for the enhanced gene- and chemotherapy to treat PC and LC [5].

With the aim of improving the accumulation of an anticancer drug (sorafenib) in LC in situ, Elsayed et al. developed sorafenib (SOR)-loaded functionalized CNTs. They produced SOR@*f*-CNTs in the form of microcapsules and examined their possible therapeutic effect both in vitro using HepG2 cells and in vivo using an HCC rat model [142]. SOR@*f*-CNTs microcapsules demonstrated delivery of sorafenib, which translated in cytotoxic effects more than two-fold higher than those of sorafenib as confirmed by Western blot and immunofluorescence analysis [142].

Since LyP-1 is a peptide binding pancreatic cancer cells, Lin et al. synthesized MWCNTs functionalized with LyP-1 residues (LyP-1@*f*-MWCNTs) and then functionalised further the achieved nanocomposite with MBD1siRNA, achieving a LyP-1@*f*-MWCNTs/MBD1siRNA complex [57]. When administered in vitro to PC cells, higher cellular uptake of LyP-1, superior gene transfection efficiency, significantly decreased viability, and proliferation and apoptosis were observed (Figure 4). Also, when injected in nude mice, the LyP-1@*f*-MWCNTs/MBD1siRNA complex significantly relieved tumour burden [57].

Lobaplatin (LOB) was loaded on PEG@CNTs, and the resulting LOB@PEG@CNTs nanocomposite was labelled with fluorescein isothiocyanate (FLUO) to visualize the cellular uptake of SOR into HepG2 cells by Yu et al. [143]. Upon administration of LOB@PEG@CNTs to LC cells nearly 80% of LOB was released at pH 5.0, characteristic of the tumour microenvironment (TME), and high cell penetrability was observed which caused 78% inhibition of HepG2 cells proliferation. Once again, this CNT-containing formulation possesses high potential in LC diagnosis, thus being a promising platform from which to develop theragnostic devices [143]. Table 7 summarizes studies on the application of CNT-based DDSs to deliver transported material to specific intracellular sites for various enhanced therapies or their combinations against PC and LC [5].

### 4.4. Discussion on the Case Studies Previously Reported

All these studies confirmed that CNTs could be excellent vectors for anticancer drugs, genetic materials, antibodies, etc., thus allowing enhanced CT, IT, and GT. Furthermore, they demonstrated great potential to develop nanocomposite sensitizers for enhanced PTT. Additionally, the functionalization of CNTs surfaces with proper ligands which can be recognized by PC and LC cells, as well as the engineering of CNTs nanocomposites structured to be capable of recognizing specific constituents of the TME or of being responsive to specific signals/conditions of the TME, could permit high accumulation at tumour sites thus allowing targeted therapy and low side effects to normal cells. Differently from CNT-based biosensors, CNT-based nanocomposites experimented for PC and LC therapy were made of both SWCNTs and MWCNTs, with a light preference for using SWCNTs. SWCNTs are more efficient in drug delivery applications than MWCNTs since they have an ultra-high surface area and good drug loading capacity. Therefore, they are preferred over MWCNTs for realizing DDSs.

The major concerns with the use of SWCNTs remain their insolubility and their higher level of impurities, which can cause toxicity. These issues can largely restrict their importance in biological and biomedical applications in vivo. Anyway, to improve the solubility of SWCNTs in aqueous solutions, pegylated nanocomposites with biocompatible properties have been widely designed.

## 5. Carbon Nanotubes for Gastric and Colorectal Cancer Therapy and Diagnosis

Gastric cancer (GC) is one of the foremost cancer-related causes of death, characterized by poor diagnosis and low survival rate, mainly because GC is often at advanced stages already at first diagnosis [112]. Similarly, colorectal cancer (CC) is a lethal disease currently treated which tumour resection, radiotherapy, and chemotherapy [112]. Anyway, due to therapeutics-induced secondary tumorigenesis, the emergence of multidrug resistant (MDR) cancer cells, and the frequent relapses the survival rate of CC-bearing patients remains low, while severe side effects detrimentally affect the quality of life of survivors during and after treatments [5]. In this regard, it is crucial to develop both enhanced diagnosis methods for GC and CC early detection, as well as to find novel enhanced therapeutic approaches to treat GC and CC and/or to improve the anticancer effects of currently existing therapeutic strategies. To these ends, CNTs could be excellent tools.

### 5.1. Carbon Nanotubes for Gastric and Colorectal Cancer Diagnosis

Early detection is essential for reducing mortality from gastrointestinal cancers (GICs), but conventional screening methods can be invasive, expensive, time consuming, and inefficient [85]. Non-invasive diagnostic approaches are promising for more efficient cancer scrutiny, including oral rinse tests assessing the oral microbiome and circulating tumour DNA (ctDNA) analysis, which have shown improved specificity in detecting gastric cancer (GC). Stool-based assays and FDA-approved blood-based tests, including Shield, are instead reforming colorectal cancer (CRC) diagnosis. Although these innovations address long-standing challenges in the early diagnosis of GC and CRC, by increasing sensitivity and patient compliance novel methods for the early diagnosis of such tumours, the outcomes of which could then depend less on the clinicians’ ability, are needed. Various CNT-based nanomaterials have been developed for theragnostic purposes due to their lightness, high surface area, considerable drug loading capacity [86], high possibility of functionalization with biological macromolecules without structural deterioration but with enhancing outcomes, and in terms of lower nanotoxicity. Anyway, a very low number of studies concerning the use of CNTs for GC and CRC early diagnosis were found in respect to the high number of works concerning the CNT-enhanced detection of other tumours. Abdolahad et al. used vertically aligned CNTs (VACNTs) fabricated on a Ni/SiO_2_/Si layer to develop an electrical cell impedance sensing biosensor (ECIS) for the detection of SW48 colon cancer cells, where VACNTs served as an adhesive and conductance mediator for the cells. A total of 30 s after the suspension of cancer cells, the impedance of the sensor changed considerably, indicating the interaction between the cells and VANTs-based biosensor [146]. The same authors reported the detection of CRC HT29 and SW480 cells using VACNTs-based electrical spectroscopy, observing that upon interaction between cells and a VACNTs-based endoscope the impedance of the sensor changed considerably, thus allowing higher efficiency and enhanced sensitivity due to the high conductivity and cells penetrability of VACNTs [147]. As previously mentioned, Ruggero et al. appended radiometal-ion chelates 1,4,7,10-tetraazacyclododecane-1,4,7,10-tetraacetic acid (DOTA) or desferrioxamine B (DFO) and tumour neovascular-targeting antibody E4G10 to SWCNTs, achieving the [^225^Ac] DOTA@SWCNTs@antiE4G10 and [^89^Zr] DFO@SWCNTs@antiE4G10 nanocomposites [148]. While [^225^Ac] DOTA@SWCNTs@antiE4G10 was administered to a murine xenograft model of human colon adenocarcinoma (LS174T) for radioimmunotherapy (RIT), with the excellent results above reported, [^89^Zr] DFO@SWCNTs@antiE4G10 was used in the same xenograft model to perform improved positron emission tomographic (PET) radio immune imaging (RII) (PET-RII) of the tumour vessels [148]. Improved signal-to-noise ratio was achieved thanks to SWCNTs, without detrimentally impacting the immunoreactivity of the targeting antibody moiety. Dynamic and longitudinal PET images showed rapid blood clearance (<1 h) and specific accumulation at tumour site [148]. PET-RII performed using the nanocomposite allowed them to visualize the pharmacokinetic (PK) profile to make possible favourable alterations of blood clearance and provided advantages for more rapid imaging [148]. Near-infrared (NIR) three-dimensional fluorescent-mediated tomography (3DFMT) (NIR-3DFMT) was used to image the LS174T tumour model, collect antibody-alone PK data, and calculate the number of copies of VE-cad epitope per cell upon a single administration of [^89^Zr]DFO@SWCNTs@antiE4G10, which was safe and well tolerated by the murine model [148].

The effective labelling of Colon-26 cells utilizing an oxidized and PEGylated SWCNT (o-SWCNT@PEG)-based over-thousand-nanometre (OTN)-near-infrared (NIR) fluorescent probe has been reported by Sekiyama et al. [149]. The authors observed a delayed concentration of probe and long-time retention in cells and an intense signal after 48 h from administration, as confirmed also by Raman imaging [149]. Additionally, since colorectal cancer cells are characterized by a prevalent K-ras mutation, thionine (TH) was immobilized on nylon 6 (PA6)-doped MWCNTs, which served as a nanosized backbone for TH electro-polymerization [150]. Detection of K-ras gene, via a multiple signal amplification strategy, caused an electrochemical transduction alteration with an impressive sensitivity of 30 fm [150]. Also, Wang et al. engineered a nanohybrid platform based on CNTs for the photoacoustic imaging of GC cells [151]. They covalently attached silica-coated gold nanorods to the surface of MWCNTs and then conjugated RGD (Arg-Gly-Asp) peptides on the produced nanostructure. The prepared probes were injected into a nude mice model with gastric cancer, and the mice were studied by photoacoustic imaging. The results showed good solubility in water, low toxicity, and good targeting of gastric cancer cells. These results suggest probable use of this type of probes for targeted drug delivery systems and PTT. Moreover, oxaliplatin (OP) was encapsulated into PEG@MWCNTs tooled with superparamagnetic iron oxide (SPIO) for magnetic resonance imaging (MRI), achieving a SPIO@PEG@MWCNTs@OP nanocomposite, by Lee et al. They observed sustained release in HCT116 cells with OP reaching 56% over 144 h, and enhanced T_2_-weighted MRI signal after intravenous administration of nanocomposite in the tumour region for early cancer diagnosis [152]. Liu and co-workers synthesized SWNTs which had five different C13/C12 isotope compositions and well-separated Raman peaks [153]. Upon functionalization with five targeting ligands, including Erbitux (Erb), Arg-Gly-Asp peptides (RGD), Rituxan (RTX), Herceptin HER, and antibody against carcino embryonic antigen (anti-CEA) Herceptin, to impart better molecular specificity the nanocomposite *f*-SWCNTs@Erb@RGD@anti-CEA@RTX@HER was achieved and used for efficient multiplexed Raman imaging of live cells [153]. Ex vivo Raman imaging experiments on tumour samples have been performed in the near-infrared (NIR) region under a single laser excitation. An almost zero interference background of imaging was obtained thanks to the definite Raman peaks due to SWCNTs over the low, smooth autofluorescence background of biological species. Raman imaging of tumour samples revealed an astonishing up-regulation of epidermal growth factor receptor (EGFR) on LS174T colon cancer cells [153]. The following Table 8 collects the found studies on the use of CNTs for early diagnosis of GC and CRC by different sensing techniques.

### 5.2. Carbon Nanotubes for Gastric and Colorectal Cancer Therapy

In the year 2009, a hydrophilic MWCNT-based drug delivery system (DDS), namely HCPT@DATEG@*f*-MWCNTs, was developed by Wu et al. by covalently binding oxidized CNTs (MWCNTs-COOH) to the antitumor agent 10-hydroxycamptothecin (HCPT) using di-amino-tri-ethylene glycol (DATEG) as the spacer between MWCNTs-COOH and HCPT [154]. Both in vitro and in vivo experiments demonstrated that HCPT@DATEG@*f*-MWCNTs exerted superior in antitumor activity than the HCPT formulation used in clinical practice due to its higher cellular uptake, longer blood circulation, and higher drug accumulation in the tumour site [154]. Tahermansouri et al. used carboxylate short (Sh) MWCNTs (Sh-MWCNTs-COOH) to synthesize amide- and spiro-derivates (Sh-MWCNTs@Amide and Sh-MWCNTs@Spiro) by adding isatin derivatives and hydrazine, respectively [155]. When these modified CNTs were administered to MKN-45 human gastric cancer cells, high cytotoxicity was observed for Sh-MWCNT@Amide, thus paving the way for its probable use in novel chemotropic strategies [155]. The same authors, the same year, further modified Sh-MWCNTs@Amide using 2-aminobenzophenone to achieve the MWCNT@quino nanocomposite. The antiproliferative effects of this were tested on MKN-45 cells, again, observing further improved cytotoxic effects [156]. Similarly, Tahermansouri and Ghobadinehad modified Sh-MWCNTs@Amide using aromatic aldehydes, producing a Sh-MWCNT@imidazole derivative which demonstrated its ability to kill 71–77% cells when administered to MKN-45 GC ones [157]. CNTs were also engineered to target gastric cancer stem cells (GCSCs) by Yoa et al. [158], identified as CD44^+^ cells, which are among the main cells responsible of GC initiation. To this end, authors coated SWCNTs with chitosan (CHI), modified the product with hyaluronic acid (HA), labelled the achieved nanocomposite with a fluorescein isothiocyanate (FITC) fluorescent probe, and stocked the final product with salinomycin (SAL), achieving the FITC-SAL@SWCNT@CHI@HA nanoplatform. Upon its administration to GCSCs and CD44^+^ cells sorted from human gastric adenocarcinoma cell line (AGSs), a significant decrease in the proportion of CD44^+^ cells, in the ability of mammosphere and colony formation, and in the growth of GCSC mammosphere were observed. Also, migration and invasion capabilities of GCSCs were significantly and selectively inhibited [158]. Experiments to investigate possible mechanisms of action revealed that the FITC-SAL@SWCNT@CHI@HA nanoplatform significantly enhanced cellular uptake of SAL via receptor-mediated endocytosis and increased its cytotoxicity to GCSCs, inducing more apoptosis of CSCs than the free drug [158]. Numerous studies on the therapeutic effectiveness of various nanoparticles loaded with platinum-based drugs against colorectal cancer (CRC) to overcome CRC-developed resistance to such chemotherapeutics, including clinically licenced cisplatin, carboplatin, and oxaliplatin, have been reported [159]. Aiming at developing CNT-based drug delivery systems for the sustained release of oxaliplatin (OP) to treat colorectum cancer cells, Wu et al. incorporated OP in PEGylated MWCNTs to create a MWCNT@PEG@OP nanocomposite and tested its anticancer effects in vitro in the HT29 cell line. A protracted and sustained release of OP was observed, with only 34% of OP released within 6 h. Additionally, a cytotoxicity dramatically higher than that of OP alone against HT29 cells was experimented, when cells were exposed to MWCNT@PEG@OP for 48 and 96 h. This trend was confirmed by experiments assessing the formation of Pt-DNA adducts and of γ-H2AX, as well as the presence of cell apoptosis [160]. Similarly, OP was encapsulated into PEG@MWCNTs tooled with superparamagnetic iron oxide (SPIO) for magnetic resonance imaging (MRI), achieving a SPIO@PEG@MWCNTs@OP nanocomposite. This produced sustained release of OP, reaching 56% over 144 h in HCT116 cells. As in the study of Wu et al., Lee and co-workers observed cytotoxic effects lower than those of OP alone when the cell viability of HCT116 cells was assessed at 12 and 24 h, while a drastic enhancement in cytotoxicity was observed at 96 h [152]. In vivo experiments demonstrated antitumor efficacy like that of OP but without adverse effects on mice, and the T_2_-weighted MRI signal was improved after intravenous administration of the nanocomposite which indicated an excellent MRI enhancement in the tumour region, which was promising for early cancer diagnosis [152]. Taghavi et al. caused synergistic apoptotic GC cell death by using a novel SWCNTs-COOH-based targeted nano delivery system (NDS) comprising Bcl-xL-specific shRNA, a very low DOX content, a modified branched poly ethylenimine (PEI 10 kDa), polyethylene glycol (PEG), and AS1411 aptamer as the nucleolin ligand to target the co-delivery system to the GC cells overexpressing nucleolin receptors [161]. The as-prepared NDS significantly inhibited the growth of nucleolin-abundant GC cells with strong cell selectivity and exerted excellent tumoricidal effects, due to the combination of anticancer gene- and chemotherapy provided by shRNAs and DOX. Furthermore, DOX concentration 58-fold lower than its IC_50_ mitigated its toxic side effects [161]. More recently, to meet the need of effective clinical means to treat peritoneal dissemination of GC, Chen et al. engineered an aptamer-siRNA chimera (Chim)/polyethyleneimine (PEI)/5-fluorouracil (5-FU)/carbon nanotube (CNT)/collagen nanocomposite (apt-siRNA Chim@PEI@5-FU@CNT@/collagen), which was capable of controlled and sustained release of 5-FU for more than 2 weeks and specifically bound to GA cells due to the aptamer-siRNA chimera, thus enabling targeted delivery of 5-FU and silencing drug-resistant oncogenes [162]. In vitro experiments showed that the nanocomposite induced apoptosis in 5-FU-resistant GC cells, while it inhibited their invasion and proliferation. Furthermore, in animal experiments it significantly inhibited the expression of mitogen-activated protein kinase (MAPK) and effectively treated peritoneal dissemination of 5-FU-resistant GC [162]. González-Domínguez et al. chemically functionalized SWCNTs with fluorescein (F), FA, and capecitabine (CAPE), a commonly used drug to fight colorectal cancer, achieving the *f-*SWCNTs@F@FA@CAPE nanocomposite which was dispersed in water with type-II nanocrystalline cellulose (II-NCC), obtaining a colloidal system (CS) [163]. CS was then tested in vitro on both normal and cancerous human colon cells (Caco-2), observing a cytotoxic activity against colorectal cancer higher than that of capecitabine alone. In addition, confocal microscopy fluorescence imaging, using cell cultures, evidenced the potential of this system to work as a potent sensor for colorectal cancer early diagnosis [163]. Recently, Jin et al. reported that using in vitro and in vivo a CpG@CNTs complex on colorectal cancer cells CpG uptake was significantly enhanced, NF-κB signal was activated, and local xenograft tumour growth and liver metastasis were suppressed. The CpG@CNTs complex cured 75% of mice and significantly prolonged their survival rate [164]. SWCNTs conjugated with antibody C225 and loaded with 7-ethyl-10-hydroxy-camptothecin (SN38) were used by Lee et al. to realize targeted therapy against EGFR over-expressed colorectal cancer cells, observing a specific binding and controlled release of SN38 and death in HCT116, HT29, and SW620 (negative control) colorectal cancer cell lines within the order of decreasing expression levels of EGFR [165]. Intraperitoneal hyperthermic chemotherapy (IPHC) is a treatment that could address the inefficiency of systemic chemotherapy in drug targeting to treat peritoneal cancers or GICs with peritoneal extensions, the overall efficacy of which is limited by prolonged anesthesia and the requirement for large amounts of drug perfusion [166]. Activated CNTs, due to their fast heating and behaviour like that of dipole antennas where optimal radiation coupling occurs when the wavelength of incident light is longer than half the length of the tube, could exert high-performance advanced IPHC [166,167]. CNTs have been experimented in IPHC in colorectal cancer (CRC) with peritoneal extensions to enhance the delivery efficiency of oxaliplatin and mitomycin C to CRC cells via infrared (IR) light stimulation [166]. Thermal ablation therapy, involving the activation of CNTs with IR radiation (700 and 1100 nm), and intracellular drug delivery are two approaches being investigated for such cancer treatment using CNTs [168]. To elevate local temperature, enhance drug absorption in malignant cells, protect adjacent tissues from intense thermal stress, and prevent excessive drug absorption in sensitive tissues, the use of hyperthermia within the ablation range (below 45 C) is recommended [169]. MWCNTs can induce hyperthermia just near tumours, thus enhancing chemotherapy uptake by cancer cells by attaching segments that can bind to tumour surfaces and thereby localizing MWCNTs to tumour nodes present in the peritoneum. By exposing the tumour nodules to chemotherapy drugs, targeting CNTs and IR light at the same time, rapid hyperthermic chemotherapy just near cell surfaces can be achieved. Additionally, during laparotomy the surgeon can identify and locate tumour nodules and direct IR light toward them, facilitating hyperthermia through CNTs transfer. The traditional IPHC approach utilizes either open or closed abdominal hyperthermic chemoperfusion [170], and intraperitoneal perfusion equipment necessitates the use of large volumes of medication which is not supported by the adult human stomach as it is unable to accommodate such quantities of fluids. Conversely, CNT-induced hyperthermia requires only 0.5 L of chemotherapy drug, eliminating the need for perfusion circuits and allowing tumour nodules to be exposed to infrared light once after being immersed in the drug/CNTs solution within the peritoneal cavity [166]. Moreover, the removal of multiple large nodules may be necessary during tumour debulking with a >10 h process, while IPHC using CNTs can significantly reduce the time required for tumour debulking and anesthesia. Employing an enhanced IPHC technique that incorporates CNTs for localized chemotherapeutic delivery and shorter treatment periods could result in improved drug uptake and improved anticancer effects [168,170].

Sundaram et al. engineered hyaluronic acid (HA)- and chlorin e6 (Ce6)-modified SWCNTs, achieving a HA@Ce6@SWCNTs nanocomposite whose anticancer effects were assessed when used for PDT to eradicate Caco-2 colon cancer cells [171]. They observed anticancer higher than those of free Ce6, mainly due to the enhanced water dispersibility conferred by HA (Figure 5).

Flow cytometry results indicate that the CNT-improved PDT enhanced both early- and late-stage apoptosis in comparison to control cells and to the use of free Ce6 [171].

Collectively, although noticeable among the research related to this field, these studies are still too few (14), as in the case of the hepatopancreatic cancer. Moreover, among these limited experiments the in vivo ones were rarely paid attention (4 out of 16, 25%). Tripisciano et al. have encapsulated irinotecan (Ir), a chemotherapeutic commonly used to treat colorectal cancer, into MWCNTs with a large inner diameter, achieving the nanoformulation Ir@MWCNTs which demonstrated high drug-filling efficiency (∼32 wt.%) for the antineoplastic agent [172]. Ir did not degrade thanks to the simple filling process carried out and its complete release occurred only in an acidic environment such as that present at the tumour site and in the lysosome upon endocytosis, forecasting a targeted behaviour and release only once inside cancer cells and not in the blood and demonstrating enhanced anticancer efficiency and low levels of toxicity to normal cells. Thus, the developed nanocarrier described by Tripisciano et al. paved the way for promising future in vitro analysis on colorectal cancer cells [172]. Ruggiero et al. appended radiometal-ion chelates 1,4,7,10-tetraazacyclododecane-1,4,7,10-tetraacetic acid (DOTA) or desferrioxamine B (DFO) as well as tumour neovascular-targeting antibody E4G10 to SWCNTs, achieving the [^225^Ac] DOTA@SWCNTs@antiE4G10 and [^89^Zr] DFO@SWCNTs@antiE4G10 nanocomposites [148]. When the nanocomposites were administered to a murine xenograft model of human colon adenocarcinoma (LS174T) for radioimmunotherapy, reduced tumour volume, improved median survival relative to controls, rapid blood clearance (<1 h), and specific accumulation at tumour site were observed and were also confirmed by positron emission tomographic (PET) radio immune imaging (RII) (PET-RII). Furthermore, near-infrared (NIR) three-dimensional fluorescent-mediated tomography (3DFMT) (NIR-3DFMT) was performed upon a single administration of [^89^Zr]DFO@SWCNTs@antiE4G10, which was safe and well tolerated by the murine model [148].

Studies, dating back more than 10 years have been summarized in Table 9, while the more recent ones have been collected in Table 10.

### 5.3. Discussion on the Case Studies Previously Reported

Case studies reported in Table 8 evidenced how the intimate structure of CNTs strongly influenced both their functions and the sensing or therapeutic techniques in which they were applied. Vertically aligned carbon nanotubes (VACNTs), unlike randomly oriented CNTs (SWCNTs and MWCNTs), are a unique class of CNTs. They are highly oriented and normal to the respective substrate on which they have been synthesized. They possess better spatial uniformity, increased surface area, greater susceptibility to functionalization, improved electrocatalytic activity, faster electron transfer, higher electrical conductivity, and higher resolution in sensing. Thank to these characteristics, some authors utilized VANCs to engineer electrical biosensors. Due to their unique highly oriented structure, VACNTs without any functionalization provided very efficient electrical-cell-impedance-sensing biosensors (ECIS-BSs) and biosensors for electrical microscopy. They demonstrated great capacity for interacting with cancer cells, fast adhesion, and high conductivity, which allowed enhanced electrical signals [146,147]. Anyway, despite these advantages, there are disadvantages, including high production costs, which limit their usage (two cases out of eight in cancer biosensing and zero in cancer therapy). The main disadvantage is that VACNTs, being oriented for definition, grow attached to a substrate. This constraint limits their usage in applications such as drug delivery or gene therapy, for which no case using VACNTs has been found (Table 9 and Table 10). While both VACNTs and CNTs can store molecules within their hollow centres, VACNTs cannot deliver the cargo in vivo since they are bound to a substrate. In this regard, when CNT-based nanocomposites are finalized to load therapeutic molecules (drug, DNA) which need to be released in the tumour site, regardless of their application as a biosensor to detect cancer or as a therapeutic device to treat tumours, SWCNTs (0.5–2.5 nm diameter) and MWCNTs (7 up to 100 nm diameter) were used (Table 8, Table 9 and Table 10). It is to be noted that, while SWCNTs were preferred when a robust cargo was uploaded since they possess drug loading capacities superior to those of MWCNTs, MWCNTs were preferred for uploading one-molecule cargo by covalent binding or for enhanced functionalization since they are more robust than SWCNTs and therefore more resistant to chemical reactions (Table 9 and Table 10). Moreover, data reported in the fourth column of Table 10 indicate that the use of CNTs in the treatment of gastrointestinal cancers led to a reduced toxicity in normal cells, a higher survival rate in animal models, a higher biocompatibility, and lower adverse effects.

## 6. Main Factors Which Hamper CNTs Translation in Clinical Practice

All nanomaterial (NM)-based drugs encounter several translational barriers and challenging steps to be approved for clinical practice, and they have been schematized in Figure 6 [173].

Anyway, several nanomaterial-based products are already on the market and used mainly to treat cancers, and even more are in clinical development. The greatest use of NMs in clinical trials encompasses a variety of drug delivery platforms, including polymeric micelles, liposomes, dendrimers, and inorganic nanoparticles, loaded with already approved drugs [173]. Despite the arsenal of nanoparticulate targeted systems currently under preclinical development or in clinical trial, liposomes were the first FDA-approved NMs (Doxil^®^/Caelyx^®^), and currently liposomes are dominant in clinical practice, followed by micellar and protein nanoparticles [30,35,36]. This because liposomes are made of biocompatible ingredients and have all the necessary features to allow the formulation of highly toxic and/or poorly soluble drugs, such as paclitaxel and amphotericin B, as well as other therapeutics [173]. Expectedly, many more NMs will progress to clinical investigation in the next few years, and again liposomal formulations represent the biggest share of NMs under clinical evaluation. The greatest clinical benefit observed by using drug-loaded nanoparticles consists of a reduction in drug toxicity, with little evidence of improved efficacy. The recently approved liposomal NM Vyxeos^®^ (daunorubicin/cytarabine liposomal formulation) demonstrated improved survival and response rates, with tolerable toxicity in phase III clinical trials in older patients with therapy-related acute myeloid leukemia (t-AML) or AML with myelodysplasia-related changes [173]. Still, very backward is instead the situation of CNTs. Cellular toxicity, genotoxicity, incompatibility with biological mediums, tendency to agglomerate, accumulation in tissues and organs, and poor degradability and consequent long-term persistence in environment and organisms are among the major downsides of CNTs application in real scenarios [5,52,54,55,77]. Also, regulatory hurdles and other factors, including high costs of production if implemented at a high scale, strongly contribute to postponing the moment when CTNs will be used in common practice [52,54]. The same problems take on even greater weight in the case of CNTs application in biomedicine, thus strongly hampering their translation in clinical uses. A wide dissertation concerning the possible CNT toxicity to organisms and environments enriched with several in vitro and in vivo studies has been collected in several informative and easy to read Tables, as well as the main tactics developed to moderate CNTs side effects, and these are available in our recent papers [5,52,54].

Mainly, surface functionalization, enhanced purity, and not-defective CNTs can reduce their toxicity, while optimization of CNTs physicochemical properties, such as surface area, diameter, and length, can play a key role in governing their biocompatibility [174]. Compared to pristine CNTs, surface-functionalized CNTs could be more water-soluble and less toxic, while surface modification using cell-specific biomolecules can promote CNTs capability to be ahead of only definite cells [175], thus reducing their systemic toxic effect and immunogenicity due to specific fast cell uptake and internalization [176]. Van der Waals interactions between bundles and high surface energies confer on CNTs a marked tendency to form aggregates, thus restraining their dispersion in most organic and inorganic solvents as well as host materials and biological fluids [1]. Bio-corona formation as surface functionalization represents an interesting approach to modify the complex surface characteristics of pristine CNTs in terms of structure and function, thus influencing toxicological properties, hydrodynamic size, aggregation, and targeting of CNTs in biological environments [177]. Cell line heterogeneity can strongly influence CNTs toxicity to cells [178]. Collectively it has been demonstrated that CNTs can promote cell death and inhibit cell proliferation by inducing overproduction of reactive oxygen species (ROS) and generating oxidative stress (OS) [179], which reduces cell adhesion capability, encourages autophagic cells death [179,180], causes membrane destabilization and DNA damage [181], and provokes pyroptosis [182] and stress endoplasmic reticulum [183]. Zhou et al. investigated the different in vitro cytotoxic effects of pristine versus functionalized CNTs (MWCNTs@COOH and MWCNTs@OH) on A549 cells, observing that pristine MWCNTs caused a higher reduction in cell viability than functionalized MWCNTs. Conversely, functionalized MWCNTs were more genotoxic than the pristine ones [184]. In vitro and in vivo results demonstrated that MWCNTs@NH_2_ and MWCNTs@COOH significantly reduced the toxic effect of pristine MWCNTs in HEK293 cells and zebra fish MWCNTs [185].

CNTs with enhanced long-term biocompatibility, degradation, and clearance in vivo are urgently needed to allow in clinic practice the exploitation of the great potential of CNTs observed in vitro and in vivo preclinical studies. Galassi et al. assessed in vivo the short- and long-term biodistribution and biocompatibility of a single-chirality DNA-encapsulated SWCNTs complex using male C57BL/6 mice at 6–12 weeks of age [186]. By means of a hyperspectral mouse imaging system, the authors observed that upon intravenous administration, after 1 h, the complex specifically accumulated in the liver and >90% of fluorescent signal attenuated by 14 days in mice [186]. Using near-infrared hyperspectral microscopy, authors found minimal long-term persistence in heart, liver, lung, kidney, and spleen [186]. Other experiments and measurements suggested short- and long-term biocompatibility inside 4 months, thus paving the way for the possible use of CNTs as preclinical research tools for in vivo drug screening and drug development without affecting acute or long-term health.

Also, a wide dissertation on the regulatory hurdles concerning the production and application of CNTs in several sectors can be found in our previous papers [52,54]. Production and application of CNT-based nanocomposites on a large scale is mainly hampered by the prohibitive costs still required for their production and purification [52,54]. Research focused on the development of automated systems for producing CNTs, possessing uniform and predictable properties as well as reduced toxicity, have recently led to the development of a carbon copilot (CARCO). CARGO is an artificial intelligence (AI)-driven platform, integrating transformer-based language models, robotic chemical vapour deposition (CVD), and data-driven machine learning models [187]. Employing CARCO, Li et al. found a new titanium–platinum bimetallic catalyst for high-density horizontally aligned carbon nanotube (HACNT) array synthesis. This catalyst outperformed traditional ones, and treasuring millions of virtual experiments an unprecedented 56% precision in synthesizing predetermined densities of HACNT arrays was achieved [187].

Collectively, the main problems which still delay CNTs clinical application in cancers treatment, comprise the not fully clarified mechanism that govern the CNTs effects in the diagnosis and therapy of cancer, the not fully unveiled mechanisms of action of CNTs on normal cells and tissues, the potential toxic effects which can occur due to their special structure, the not fully clarified mechanisms of CNT-induced toxicity, and the low biodegradability of raw CNTs due to their hydrophobic properties which prevent enzymes from approaching them, thus impeding enzymatic degradation [5]. Promotion of CNTs translatability in clinic practice can only be sped up by more labour to explore the best functional molecules for CNTs modification to improve their biocompatibility, to avoid the toxic and risk factors, and to enhance their solubility and create defect sites which offer desired binding sites for enzymes and promoting enzymatic degradation [188].

### 6.1. Approaches to Reduce Possible CNT Toxicity

CNTs can accumulate in the environment and human bodies, thus possessing potential severe hazardous outcomes which can also be caused from the CNT-induced activation of toxic systems in the tissues of organisms that encounter them [55,189,190,191,192,193,194]. Several scientists have therefore studied strategies to minimize these events. Interventions are mainly based on CNT surface modifications and functionalization [55,189,190,191,192,193,194]. Polyethylene glycol (PEG), C1q recombinant globular proteins, and biocompatible ingredients or molecules which can improve the solubility and dispersibility of CNTs in biological fluids have been used to modify CNTs’ surfaces. Toxicity of CNTs can also be attributable to their effects on ROS production. In this regard, to reduce MWCNTs’ capacity to increase ROS production, thus inducing oxidative stress (OS), oxidative damage, inflammation, and immune-toxic effects, antioxidant natural molecules, such as curcumin or quercetin, have been experimented on [195,196,197]. Additionally, to effect cell uptake the CNT surface was functionalized with -COOH and -OH groups [198,199]. Moreover, since an effective purification of CNTs is essential to reduce their toxicity, often due to residual metals or catalysts which are noxious to living organisms, several advanced purifications workups have been experimented to obtain highly pure, non-defective, and less toxic CNTs [200]. Furthermore, more biodegradable CNTs over time have been developed to limit their persistence in the blood and reduce tissue toxicity [201]. Table 11 collects some important strategies proposed so far to reduce the possible dangerous outcomes that could come from extensive exposure to CNTs, as reported in our recent article [5].

### 6.2. CNTs Technologies Closest to the Clinical Application

However, despite all of the promises associated with the application of CNTs in biomedicine, the most important concerns continuously raised by scientists are based on CNT nanotoxicology and the environmental effects of CNTs, mostly due to their non-biodegradable state, which justify the lack of widespread FDA approval for CNTs [56].

The CNT technologies closest to clinical application regard those finalized for the preparation of CNT-based nanocomposites for cancer diagnosis and imaging. At the moment, no clinical practice has ever been reported concerning the application of CNTs as DDSs in cancer treatments and diagnosis, while there are some clinical trial cases that assessed the employment of carbon nanoparticles (CNPs) in lymphatic monitoring during colorectal cancer surgeries, lymph nodes collection in advanced gastric cancer, and lymph node biopsy of papillary thyroid carcinoma, confirming the great potential and promising future of CNTs in cancer theragnostic [202,203,204]. Appendix A summarize two other clinical trials carried out to evaluate the possible applications of CNT-based nanocomposites in anticancer treatments and cancer diagnosis [205,206,207]. The goal of the first study (Clinical Trial NCT01773850, United States, completed in 2019) was to compare the radiologist confidence level in evaluating patients with known breast lesions using a carbon nanotube X-ray-based stationary breast tomosynthesis imaging device. The comparison was made against conventional mammography acquired as a part of a standard clinical workup. One hundred patients, who had to have a clinical surgical breast biopsy, were recruited for the study. The second clinical trial was performed to study the feasibility of a novel method in oncology, based on breath analysis with custom-designed nano sensors based on organically functionalized gold nanoparticles and CNTs. The trial enrolled a total of 1000 volunteers. Results from this study evidenced that CNT-based sensors could be non-invasive screening tool for GC and related precancerous lesions, as well as for surveillance of the latter [206].

Other clinical trials regarded the application of CNTs to treat dental caries [208], Parkinson’s [209], and cervical pain [210].

## 7. Authors Considerations

According to Figure 7, created using data collected in our last paper on CNTs [5] and in this review, studies on the use of CNTs to treat the most lethal and perfidious cancers (CRC, PC, and LC (40)) represent only 37% of those regarding CNT-based treatments for most existing tumours (107). Specifically, using tables present in our recent review [5], reporting case studies about the CNT-based devices created for the diagnosis and therapy of the most reported tumours and tables reported in this study, and collecting studies about CNT-based nanocomposites applied for the diagnosis and therapy of CRC, PC, and LC, we obtained numerical data which were used to construct the bars graph using Microsoft excel 365 software.

This percentage even worsens if clinically approved anticancer nanomaterials are considered, where only 22% of those on the market are finalized to treat PC, metastatic PC, pancreatic adenocarcinoma, and hepatocellular cancer/liver cancer (HCC/LC) (Table 12 [30,35,36], Figure 8). Table 12 is a reproduction from an open access article published under a Creative Commons CC BY 4.0 license [68], which permits the free download, distribution, and reuse, provided that the author and preprint are cited in any reuse. The bar graph in Figure 8 has been constructed using numerical data obtained by counting the case studies reported in Table 12 for each type of tumour.

Additionally, despite gastric adenocarcinoma (GAC) not being among the five most lethal tumours, it carries a very poor prognosis, mainly due to the late stage of diagnosis [78]. Despite this, studies on the use of CNT-based nanocomposites for its early diagnosis represent only 2.9% of those regarding the diagnosis of tumours in the complex (Figure 9). The bar graph in Figure 9 was constructed using data collected in our last paper on CNTs [5] and in this review. Specifically, using tables present in our recent review, reporting case studies about the CNT-based devices engineered for the diagnosis and therapy of the most reported tumours, and tables reported in this study collecting case studies about CNT-based nanocomposites manufactured for the diagnosis and therapy of PC, LC, GC, and CRC, we obtained numerical data which were used to construct the bar graph by means of Microsoft excel 365 software. Search strategies and eligibility criteria used to construct tables from which we take data to create the bar graph of Figure 9 were the same as those described previously at the end of Introduction.

Collectively, the studies concerning the application of CNT-based nanocomposites to treat and detect PC, LC, GC, CRC, and RC account for 51% of those concerning the most relevant tumours (Figure 8 and Figure 9). Anyway, concerning only treatments, they represent only 45%, while the studies on CNT-based nanomaterials for PC, LC, GC, CRC, and RC diagnosis reach 71%, thus evidencing a considerable discrepancy between the researchers’ interest in the diagnosis of these tumours and their only treatment. This can be justified by thinking that often CNT-based sensors for diagnosis can simultaneously realize cancer treatment. Anyway, readers can note that concerning studies reporting on treatment and diagnosis of the most relevant tumours, including those considered in this review, using CNTs, those on diagnosis represent only 24%, while those on the diagnosis of only PC, LC, GC, CRC, and RC reach 33%. Therefore, it is as if the interest for the diagnosis of these specific tumours is greater than that for tumours in the complex. If from the number of studies on the most relevant tumours in the complex then those on LC, PC, GC, CRC, and RC are extrapolated, then the studies concerning the diagnosis using CNTs for tumours which are other of those considered in this paper represent only 14% of those found in total, as if the positive impacts of introducing CNTs in the diagnosis of tumours other than LC, PC, GC, CRC, and RC were less felt. Concerning the use of CNTs to treat and detect cancers considered in this paper, GC was the least studied, with only 17% of works concerning its treatment and 4% its diagnosis using CNTs. Experiments on the use of CNTs to treat CRC were the most numerous (33%), followed by PC and LC (25%), while the most numerous concerning diagnosis regarded LC (42%), followed by CRC (29%) and PC (25%).

## 8. Conclusions and Recommendations

Nowadays, it is generally recognized that early diagnosis is crucial to improving the chances of effective treatment and cure for all tumours. Detecting cancer in its early stages can mean a greater chance of successful treatment and a better quality of life for patients. Prevention policies and continuous population screening are in place for most tumours and several diagnosis steps and methods exist, such as blood and urine tests to detect specific markers indicating the presence of a possible neoplasm; genetic tests; diagnostic imaging, including X-ray, ultrasound, computed tomography (CT), and magnetic resonance imaging (MRI); and biopsy and histopathological analyses which are more efficient than anticancer drugs or anticancer treatments for tumours when discovered. Anyway, new more efficient options to both treat and detect tumours are urgently needed to improve patients’ conditions, quality of life, compliance with therapies, and survival rate.

As already evidenced in our previous paper on the use of CNTs to detect and treat the most known cancers, this umbrella review has evidenced more in-depth the great advantages which could be derived from applying CNTs to engineer CNT-based devices for the diagnosis and therapy of specific tumours, including the most lethal and perfidious cancers (CRC, PC, and LC) and the cancers having a low survival rate due to too late diagnosis or the lack of efficient instruments for early diagnosis (GC). Used alone or in combination with available therapeutic strategies, such as photothermal, photodynamic, drug targeting, and gene, immune, and chemotherapies, CNTs have shown notable results in laboratory settings in allowing the manufacturing of therapeutics and theragnostics with enhanced anticancer efficiency and reduced toxic effects compared to traditional chemotherapeutics.

Anyway, data reported in this review have also evidenced that despite PC, LC, and CRC are reported as the three among the five tumours (60%) considered the most aggressive cancers affecting humans, GC has a very low survival rate due to a too late diagnosis, and the potential for CNTs to ameliorate this scenario has been confirmed, studies on the use of CNTs to treat them represent only the 37% of those regarding the CNT-based treatment of tumours in the complex. Still too limited and often obsolete experimentation on the possible use of CNTs for the diagnosis and therapy of such tumours is a great concern which needs a solution. The scope of this review was to sensitize researchers to pay more attention to the great advantage that the association of CNTs with anticancer drugs, anticancer treatments, and methods for cancer diagnosis could have. The more efficient treatments and early diagnosis of all cancers, which could derive from extensive and safe applications of carbon nanotubes, could significantly ameliorate patients’ survival rate and quality of life by reducing doses and times of treatments, and therefore their side effects.

## Figures and Tables

**Figure 1 ijms-26-09201-f001:**
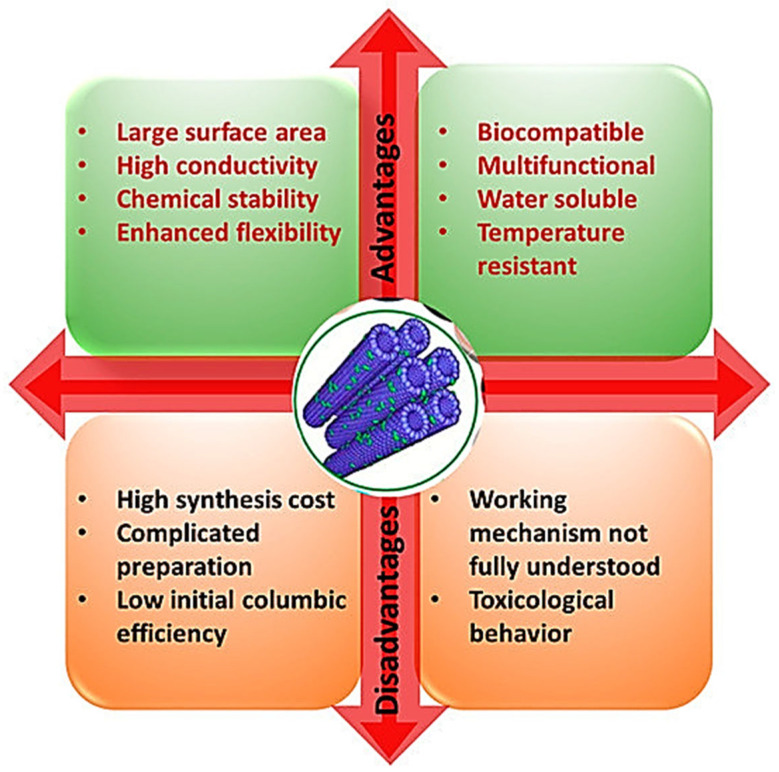
Advantages and disadvantages of CNTs [1]. The image has been reproduced from an open access article published under a Creative Commons CC BY 4.0 license [68], which permits the free download, distribution, and reuse, provided that the author and preprint are cited in any reuse.

**Figure 2 ijms-26-09201-f002:**
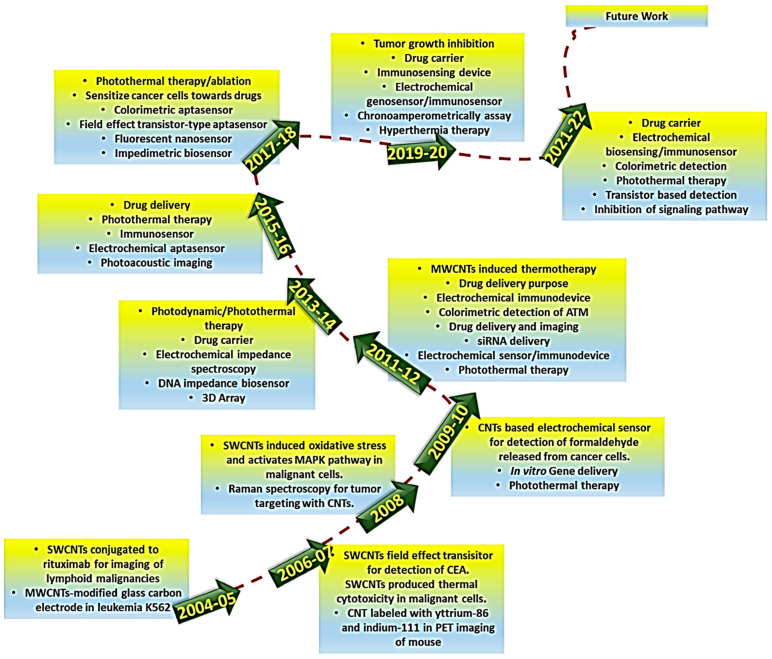
A roadmap for the evolution of the use of carbon nanotubes in cancer targeting and diagnosis [1]. The image has been reproduced by an open access article published under a Creative Commons CC BY 4.0 license [68], which permits the free download, distribution, and reuse, provided that the author and preprint are cited in any reuse.

**Figure 3 ijms-26-09201-f003:**
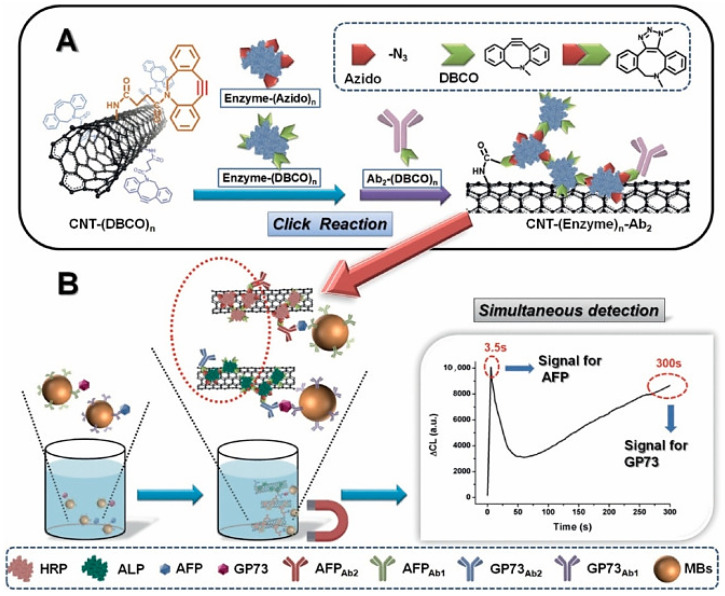
(**A**) Schematic illustration of the multifunctional immune nano sensor preparation using DBCO-PEG5-NHS ester (DBCO) surface-modified CNTs. Alkaline phosphatase (ALP) and horseradish peroxidase (HRP) were linked with the reactive groups of azido and DBCO, correspondingly. After modifying the antibody of the Golgi protein 73 and α-fetoprotein with DBCO, they were coupled to CNTs. (**B**) Simultaneous sensing of Golgi protein 73 and α-fetoprotein as hepatocellular carcinoma biomarkers. nano sensors formed double antibody sandwich complexes for concomitant detection of both biomarkers [102]. Copyright by Elsevier (2021). Reproduced under license order number 501991570 and license number 6040780373949, dated 2 June 2025 [103].

**Figure 4 ijms-26-09201-f004:**
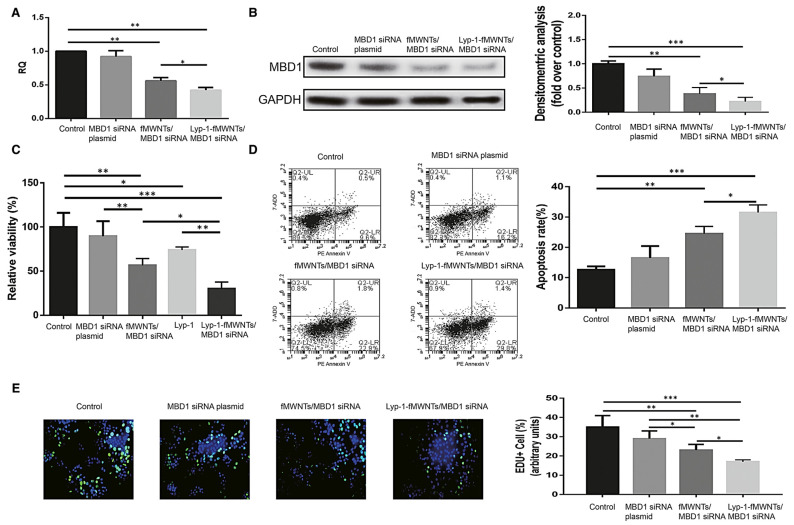
(**A**) Treatment of pancreatic ductal adenocarcinoma (PDAC) with LyP-1@ MWCNT@MBD1siRNA with promotion of cell apoptosis and inhibition of cell viability and cancer proliferation. (**B**) MDB1 quantitation in a BxPC-3 human pancreatic cancer cell line after RNA interference. (**C**) BxPC-3 viability after RNA interference. (**D**) Cytometry plots of BxPC-3 cells after RNA interference. (**E**) EdU (5-ethynyl-2′-deoxyuridine)-positive BxPC-3 cells after RNA interference. Blue signal represents DAPI nuclear staining [57]. The image has been reproduced by an open access article published under a Creative Commons CC BY 4.0 license [68], which permits the free download, distribution, and reuse, provided that the author and preprint are cited in any reuse; *, **, *** indicate statistical significance.

**Figure 5 ijms-26-09201-f005:**
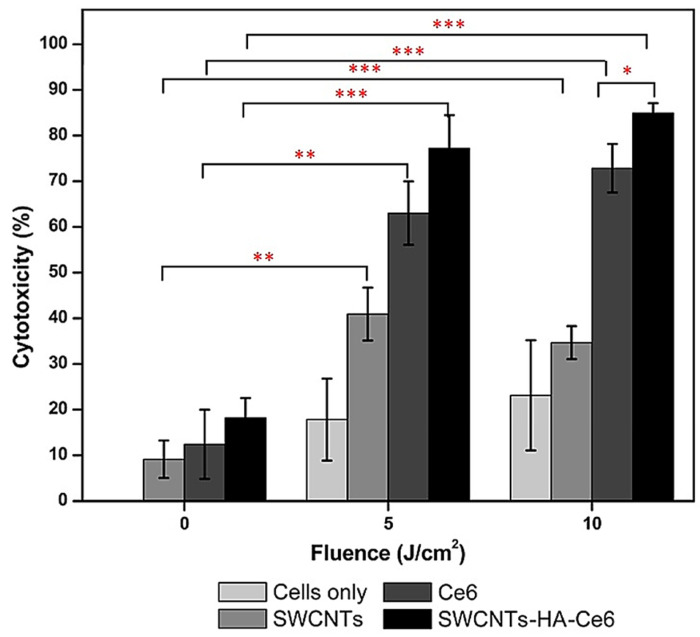
Cytotoxicity of SWCNTs, Ce6, and SWCNTs@HA@Ce6 on Caco-2 cells determined by LDH assay. Significance is shown as * *p* < 0.05; ** *p* < 0.01; *** *p* < 0.001 [171]. The image has been reproduced by an open access article published under a Creative Commons CC BY 4.0 license [68], which permits the free download, distribution, and reuse, provided that the author and preprint are cited in any reuse.

**Figure 6 ijms-26-09201-f006:**
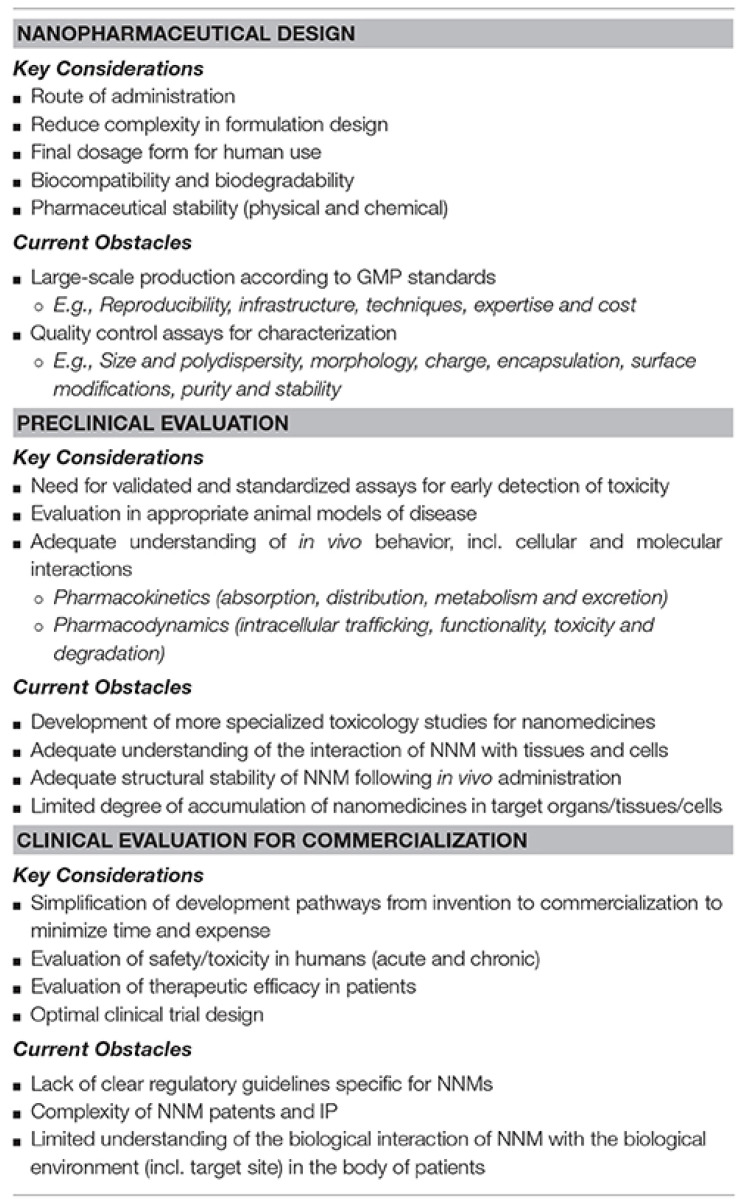
Trends and challenges in the clinical translation of nanoparticulate nanomedicines. The image has been reproduced by an open access article [173] published under a Creative Commons CC BY 4.0 license [68], which permits the free download, distribution, and reuse, provided that the author and preprint are cited in any reuse.

**Figure 7 ijms-26-09201-f007:**
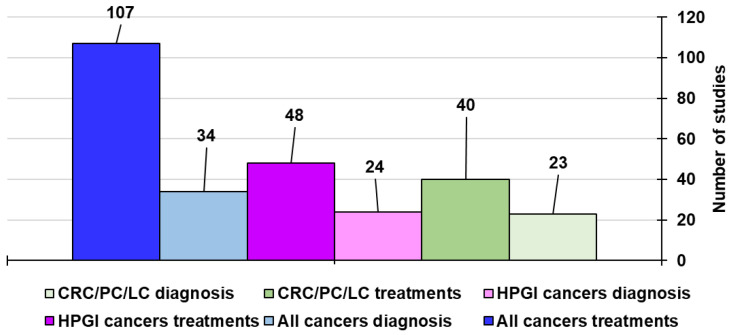
Comparison between the number of studies on the application of carbon nanotubes (CNTs) in the treatment and diagnosis of various types of cancers, studies about the specific treatment and diagnosis of hepatopancreatic and gastrointestinal (HPGI) ones, and studies about the treatment of 3 of the 5 cancers considered the most lethal worldwide. This bar graph has been elaborated based on data collected in this review and in our recent work [5].

**Figure 8 ijms-26-09201-f008:**
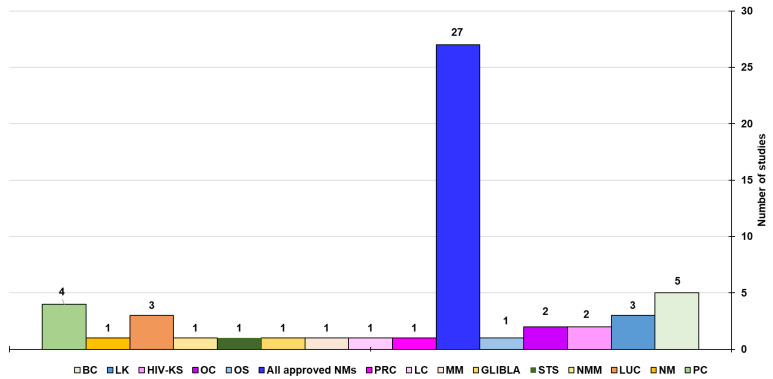
Nanomaterial-based treatments for various types of cancers clinically approved and on the market. LUC = lung cancer; BC = breast cancer; PC = pancreatic cancer; LC = liver cancer; OC = ovarian cancer; OS = osteosarcoma; HIV-KS = HIV-associated Kaposi’s sarcoma; MM = multiple myeloma; LK = leukemia; OS = osteosarcoma; PRC = prostate cancer; NMM = non-myeloid malignancies; STS = soft-tissue sarcoma; NM = neoplastic meningitis. The data to build bar graphs have been extracted from Table 12.

**Figure 9 ijms-26-09201-f009:**
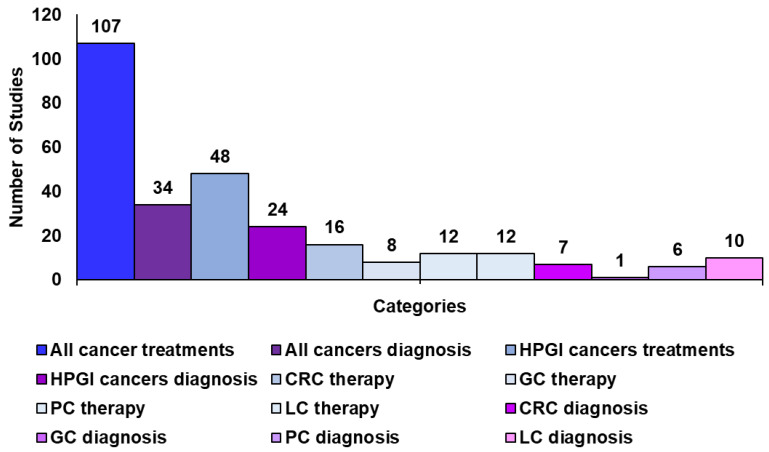
Comparison between the number of studies on the application of carbon nanotubes (CNTs) in the treatment and diagnosis of various types of cancers, on the treatment and diagnosis of hepatopancreatic and gastrointestinal (HPGI) cancers, and of studies about the specific treatment of CRC, PC, LC, and GC.

**Table 1 ijms-26-09201-t001:** Comparison between the in vitro/in vivo toxicity of CNTs and that of various types of NPs [77].

NPs (nm)	Concentration/T_E_	Species/Cell Culture	Assay	Result
Al_2_O_3_ (8–12)	1–10 µM/24 h	HBMVECs	MTT/DHE	↓ Cell viability, ↓ mitochondrial function, ↑ OS Alter proteins expression of the BBB
Al_2_O_3_ (50–80)	10–400 µg/mL/24 h	Mammalian cells	EZ4U	No significant toxic effect on cell viability
Al_2_O_3_ (160)	25–40 µg/mL/12 h	HMSC	MTT	↓ Cell viability
Al_2_O_3_ (30–40)	500–2000 mg/kg/72 h	Rat blood cells	Comet, micronucleus	Dose-dependent genotoxicity
Al_2_O_3_ (50)	0–5000 µg/mL/2 h	MLCL	Comet	DNA damage
CuO (50)	10–50 µg/mL/24 h	HLECs	MTT/LDH	↓ Cell viability, ↑ LDH, ↑ lipid peroxidation
MWCNTs (20)	0.002–0.2 µg/mL/4 days	LCCs	MTT	↓ Cell viability
SWCNT (800)	0–400 µg/mL/10 days	HACECs, NHBECs	Clonogenic	Cell death
SWCNTs (10–30)	40 and 200 µg/mouse, 1 mg/mouse, 90 days	in vivo	Commercial kits	↑ LDH, ↑ AST, ↑ ALT
Fullerenes (178)	1 ng/mL/80 days	CHO, HELA, HEK293	Micronucleus test	DNA strand breakage, chromosomal damage
Silica (15–46)	10–100 µg/mL/48 h	HBACCs	DCFH-DA/Commercial kit	↑ ROS, ↑ LDH, ↑ Malondialdehyde
Silica (43)	25–200 µg/mL/3–24 h	HepG2	DCFH-DA/TEBCI	↑ ROS, MD, OS
Ag (15–100)	10–50 µg/mL/24 h	BRL 3A	LDH, MTT, Glutathione, DCFH-DA	↓ Cell viability, ↑ LDH, ↑ ROS
Ag (30–50)	0–20 µg/mL/24 h	HACs	MTT, DCFH-DA	↓ Cell viability, ↑ ROS
Ag (20–40)	---	HLKCs	WST-1, LDH	↓ Cell viability, ↑ LDH
ZnO (50–70)	11.5 µg/mL/24 h	HCCCs	ELISA, flow cytometry	↑ OS, ↓ Cell viability, inflammatory biomarkers
ZnO (307–419)	10–100 µg/mL/24–48 h	HEp-2	Comet micronucleus test, MTT	DNA damage; ↓ cell viability
ZnO (30–70)	14–20 µg/mL/12 h	in vivo	MTT, Comet, DCFH-DA	↓ Cell viability; DNA damage; ROS, apoptosis
ZnO (50)	0–100 µg/mL/24 h	HEK 293	MTT; Comet	DNA damage, ↓ cell viability, OS, MD
ZnO (<20)	100 µg/mL	HBECCs	-	↓ Cell viability, ↑ OS, ↑ LDH release
Fe_2_O_3_ (30)	25–200 µg/mL/2 h	MMCs	MTT	↓ Cell viability
Fe_2_O_3_ (100–150)	0.1 mg/mL/7 days	HMs	MTS	↓ Cell viability
Fe_2_O_3_ (13.8)	123.52 µg/mL/12 h	HPCCs	MTT	↓ Cell viability
Fe_2_O_3_ (20)	0.1 mg/mL/2 days	RMSCs	MTS	↓ Cell viability
TiO_2_ (160)	1800 µg/mouse/10 days	in vivo	Comet, micronucleus test	DNA damage, genotoxicity
TiO_2_ (<100)	10–50 µg/mL/6–24 h	HLCs	ELISA, Trypan blue, DCFH-DA	↑ OS, DNA adduct formation, ↑ cytotoxicity

T_E_ = exposure time; HLECs = human lung epithelial cells; LCCs = lung cancer cells; HBACCs = human bronchoalveolar carcinoma cells; HPCCs = hepatocellular carcinoma cells; OS = oxidative stress, MD = mitochondrial damage; HAC = human alveolar cell line; HLKC; human leukemia cell line; HCCCs = human colon carcinoma cells; HEp-2 = human cervix carcinoma cell line; HEK 293 = human hepatocytes cell line; HBECs = human bronchial epithelial cells; MMCs = murine macrophage cells; HMs = human macrophages; RMSCs = rat mesenchymal stem cells; HLCs = human lung cells; HBMVECs = human brain microvascular endothelial cells; DHE = dihydroethidium; BBB = blood–brain barrier; HMSCs = human mesenchymal stem cells; MLCL = mouse lymphoma cell line; LDH = lactate dehydrogenase; MWCNTs = multi-walled carbon nanotubes; SWCNTs = single-walled carbon nanotubes; HACEC = human alveolar carcinoma epithelial cell line; NHBEC = normal human bronchial epithelial cell line; AST = aspartate transaminase; ALT = alanine transaminase; CHO = Chinese hamster ovary cells; HELAs = human epidermoid-like carcinoma cells; HEKs = human embryonic kidney cells; DCFH-DA = Dichlorodihydrofluorescein diacetate; ROS = reactive oxygen species; BRL 3A = Buffalo rat liver cells. ↑ = increase and ↓ = decrease; TEBCI = tetraethyl-benzimidazo-lylcarbo-cyanide iodine.

**Table 2 ijms-26-09201-t002:** CNTs-based methods for detection of cancer biomarkers.

CNT-Based Sensor	Biomarker/Effectiveness	Cancer	Refs.
MWCNTs	CA19-9	PC	[94]
MWCNTs	GdGTP in DNA	PC	[98]
SWCNTs@immune sensor	CA19-9	PC	[97]
Au@CNTs	AFP, AFP-L3, APT	LC	[101]
DBCO-PEG5-NHS ester@CNTs	GP73, α-FTP	HCC	[102]
Collagen@patterned CNTs	Detection of CCs versus NCs	LC	[104]
Zein NPs@MWCNTs	H_2_O_2_ monitoring HepG2 cells	HepG2	[105]
MWCNTs (NBS)	⬆ BA, WLRD miR-21, ⬇⬇⬇ LOD	PC	[106]

CA19-9 = carbohydrate antigen 19-9; GdGTP = guanine and deoxyguanine triphosphate in the DNA; AFP = α-fetoprotein; AFP-L3 = α-fetoprotein variants; APT = abnormal prothrombin; GP73 = Golgi protein 73; α-FTT = α-fetoprotein; LC = liver cancer; PC = pancreas cancer; HCC = hepatocellular carcinoma; CCs = cancer cells; NCs = normal cells; BA = biosensing ability; WLRD = wide linear range for detecting; LOD = limit of detection; ⬆ = enhancing, enhanced, improving, improved, augmented, high, and higher; ⬇⬇⬇ = strongly low, lower, decreased, reduced, and reduction.

**Table 3 ijms-26-09201-t003:** Carbon nanotube-based cancer detection techniques.

CNTs	Cell Line/Biomarkers	Linear Range	LOD	Techniques	Ref.
MWCNTs	AFP (LC)	0.02–2.0 ng/mL	8.0 pg/mL	Immune sensing	[107]
MWCNTs	AFP (LC)	0.1–15.0 and 15.0–200.0 ng/mL	0.08 ng/mL	Immune sensing	[108]
MWCNTs	CA 19-9 (PC)	12.5–270.0 U/mL	8.3 U/mL	Immune sensing	[109]
CNTs	AFP (LC)	1–55 ng/mL	0.6 ng/mL	Immune sensing	[110]
MWCNTs	CA19-9 (PC)	0–1000 U/mL	N.R.	Electrochemical	[94]
CNTs	GP73 (HCC)	0–80 ng/mL	58.1 pg/mL	Immune sensing	[102]
CNTs	AFP (LC)	0–64 ng/mL	47.1 pg/mL	Immune sensing	[102]
CNTs	HepG2 (LC)	10–10^5^ cells/mL	5 cells/mL	Electrochemical	[111]

LC = Liver cancer; PC = pancreas cancer; HCC = hepatocellular carcinoma; CCs = cancer cells; N.R. = not reported.

**Table 4 ijms-26-09201-t004:** Studies on CNTs applications in pancreas cancer treatment before 2020. Underlined abbreviations indicate the functionalizing molecules.

CNTs-Based NC	Highlights	Refs.
*f*-HSA@MWCNTs	HAS-MWCNTs accumulates in PCCs for laser irradiation PTT	[123]
EGF@SWCNTs	⬆ Rapid uptake of nanocomposite	[124]
*f*-ETO@ SWCNTs	Synergic effect of SWCNTs and ETO	[125]
*f*-PEI@SWCNTs	PEI@SWCNT pass both cytoplasmic and nuclear membranes	[126]
*f*-PEG@MWCNTs	Cause apoptosis pathways to activate through mitochondrial deficiency	[127]
*f*-GA@SWCNTs	⬆ GA effects by usage of CNTs	[128]
*si*RNA@SWCNTs	⬆ siRNA targeted delivery, ⬆ transfection, ⬆ gene therapy effects	[129]
*f*-PEG@*o*MWCNTs	⬆ Cell toxicity, ⬆ PTT, ⬆ TVR in animal group (PEG-O-CNTs)	[130]
*f*-PEG@GEM@SWCNTs	⬇ Metastatic lymph nodes in BxPC-3–B/c	[131]
*f*-HIF-1α/siRNA@SWCNTs	⬆ Transfection in PCCs, ⬆ RNA response, ⬇⬇⬇ TG, ⬇ harm to NCs	[132]

NC = nanocomposite; *f* = functionalized; *o* = oxidized; HAS = human serum albumin; GA = gambogic acid; EGF = epidermal growth factor; PEG = polyethylene glycol; PEI = polyethyleneimine; ETO = etoposide; PCCs = pancreatic cells; PTT = photothermal therapy; ⬆ = improved, enhanced; TVR = tumour volume reduction; GEM = gemcitabine; B/c = Balb/c mouse; HIF-1α = hypoxia-inducible factor-1 alpha; TG = tumour growth; ⬇ = low; ⬇⬇⬇ = suppression.

**Table 5 ijms-26-09201-t005:** Applications of CNTs in LC treatment. Underlined abbreviations indicate the functionalizing molecules.

CNTs-Based NC	Highlights	Refs.
HSA@MWCNTs	⬆ Apoptosis by PTT in HepG2 cancer cells than in normal ones	[137]
CHI@FA@SWCNTs@DOX	⬆ DL (%), ⬆ AE, ⬆ targeting capacity, ⬆ biocompatibility, ⬆ WS	[138]
RuPOP@MWCNTs	⬆ Cellular uptake, ⬆ AE, ⬆ apoptosis induction under X-ray	[139]
CoMoCAT^®^-SWCNTs@PL@PEG-NH_2_	⬆ Tumour growth inhibition by apoptotic death under TAT	[140]
as-ODNs@PAMAM-NH_2_@*f*-MWCNTs	⬆⬆⬆ Gene delivery efficiency	[136]
DOX@SWCNTs/DOX@HBA@SWCNTs	pH-dependent DR, max DR at pH = 5.5, ⬆ cytotoxic effects	[141]
PEG@o-MWCNTs	⬇ cytotoxicity in HepG2, ⬇⬇⬇ tumour size under PTT in vivo	[130]

NC = nanocomposite; WS = water dispersibility; DL (%) = drug loading percentage; TAT = thermoacoustic therapy; as-ODNs = antisense c-myc oligonucleotides; PAMAM = poly-amidoamine; HAS = human serum albumin; CHI = chitosan; FA = folic acid; DOX = doxorubicin; RuPOP = ruthenium polypridyl complex; *f* = functionalized; PL = phospholipid; PEG = polyethylene glycol; CoMoCAT^®^ = synthetic method to prepare high purity SWCNTs with specific chirality and narrow distributions of tube diameters; ⬆ = high, higher, improved, and enhanced; ⬇⬇⬇ = strong reduction; DR = drug release; MR = maximum release; AE = anticancer efficiency; ⬇ = lower; HBA = hydrazine-benzoic acid; ⬆⬆⬆= extremely higher.

**Table 6 ijms-26-09201-t006:** CNT-based strategies for liver and pancreas cancer treatment. Extracted from Alfei et al. [5].

CNT-Based Formulation	Treatment Approach	Cancer Type	Refs.
LyP-1@siRNA@MWCNTs	Delivery of siRNA	Pancreatic cancer	[57]
*f*-CNTs	Sorafenib delivery	HepG2 cell line	[142]
PEG@CNTs	Lobaplatin delivery	HepG2 cell line	[143]

IGRF = insulin-like growth factor receptor; PEG = polyethylene glycol; PTT = photothermal therapy; *f* = functionalized; LyP-1 = cyclic 9-amino-acids homing peptide that specifically binds to p32 receptors overexpressed in tumour cells.

**Table 7 ijms-26-09201-t007:** Application of functionalised CNTs and CNT-based DDSs capable to accumulate and/or deliver transported material to specific intracellular sites for enhanced phototherapy (PTT), immunotherapy (IT), and chemotherapy (CT), or their combinations against PC and LC. Extracted from Alfei et al., 2025 [5].

TT	DDS	TM	Model	Effectiveness	Refs.
Cytoplasm	SWCNT@CY7@IGF-1Ra	PTT, IT	ASPC-1, BXPC-3, PANC-1 SW1990 *	BS, LT, PTTT, ⬆ BW, ⬆ SR	[99]
Macrophages	*ws*-MWCNTs@COOH	IT	H22 HCCs	⬆ WS, ⬆ CSA, ⬆ CP, ⬆ MA	[144]
140TV	DOX/CD-CNT, CUR/CD-CNT	PTT + CT	Hepatocellular	⬆ DEE and achieved sustained release of both drugs	[145]

TT = therapeutic target; TM = therapeutic modality; DDS = drug delivery system; IT = immunotherapy; * pancreatic cancer cells; IGF-1R = insulin-like growth factor-1Ra; SWNT = single-wall carbon nanotube; PTT = photothermal therapy; ⬆ = high, higher, improved, and enhanced; CY7 = cyanine 7 dye; PTTT = precise tumour target therapy; BW = body weight; SR = survival rate; WS = water soluble; BS = bio stable; LT = low toxic; HCCs = hepatocarcinoma cells; CSA = component system activation; CP = cytokines production; MA = macrophages activation; *ws* = water soluble; TV = tumour vasculature; CT = chemotherapy; CUR = curcumin; DOX = doxorubicin; CD = candesartan; DEE = drug entrapment efficiency.

**Table 8 ijms-26-09201-t008:** Studies on the use of CNTs for early diagnosis of GC and CRC, by different sensing techniques.

ST	CNTs	Model	Results	Refs.
PET-RII *	[^89^Zr] DOTA@SWCNTs@antiE4G10	LS174T **	**⬆** STNR by only SA, WT	[148]
RI	*f*-SWCNTs@Erb@RGD@anti-CEA@RTX@HER	LS174T **	Zero interfering, ⬆ EGFR, MCOI	[153]
ECIS-BS	VACNTs	SW48 CRCCs	⬆ Adhesivity, ⬆ conductance, ⬆ cells interaction	[146]
EM	VANCTs	HT29, SW480	⬆ Efficiency, ⬆ sensitivity, ⬆ conductivity ⬆ CP	[147]
FI, RI	o-SWCNT@PEG	Colon-26	DC, ⬆ RT in cells, ⬆ FS after 48 h	[149]
ECBS	PA6@MWCNT@SH@	CRCCs	⬆ K-ras GD, MSAS, ⬆ sensitivity (30 fm)	[150]
PAI	Si/Au@MWCNTs@RDG	GCCs	⬆ WS, ⬇ toxicity, ⬆ TTC	[151]
MRI	SPIO@PEG@MWCNTs@OP	HCT116	⬆ T_2_-weighted MRI signal after IVA	[152]

MRI = magnetic resonance imaging; ST = sensing technique; EM = electrical microscopy; PAI = photoacoustic imaging; RI = Raman imaging; FI = fluorescence imaging; ⬆ enhanced, improved, high, and higher; ⬇ low, lower, decreased, and reduced; Erb = erbitux; RGD = Arg-Gly-Asp peptides; RTX = Rituxan; HER = Herceptin; MCOI = multi-colour optical imaging; IVA = intravenous administration; GCCs = gastric cancer cells; CRCCs = colorectal cancer cells; WS = water solubility; TTC = tumour target capacity; SH = thionine; PA6 = nylon 6; MSAS = multiple signal amplification strategy; CP = cell penetrability; anti-CEA = antibody against carcino embryonic antigen; EGFR = epidermal growth factor receptor; ECIS-BS = electrical cell impedance sensing biosensor; ECBSs = electrochemical biosensors; RT = retention time; DC = delayed concentration; FS = fluorescence signal; GD = gene detection; PET = positron emission tomographic; RII = radio immune imaging; NIR = near-infrared; 3DFMT = three-dimensional fluorescent-mediated tomography; * associated with NIR-3DFMT; SPIO = super-magnetic iron oxide; OP = oxaliplatin; STNR = signal-to-noise ratio; SA = single administration; WT = well tolerated; ** colorectal adenocarcinoma cell.

**Table 9 ijms-26-09201-t009:** Studies on the CNTs application in the colon and gastric cancer treatment.

CNTs-Based NC	Highlights	Refs.
HCPT@DATEG@*f*-MWCNTs	⬆ Antitumor activity in MKN-28 cells *	[154]
[^225^Ac]/[^89^Zr] DOTA@SWCNTs@antiE4G10	⬇ TM, ⬆ MS, ⬆ TS accumulation, RC (CRC)	[148]
Sh-MWCNTs@Amide	⬆ Toxicity to MKN-45	[155]
Sh-MWCNTs@Quino	⬆ Toxicity to MKN-45	[156]
Sh-MWCNTs@Imidazole	MKN-45 cells viability was lowered by 71–77%	[157]
FITC-SAL@SWCNT@CHI@HA	Significant decrease in mammosphere (GC)	[158]
Ir@MWCNTs	⬆ DL, ⬆ inner diameter tube, pH sensitive (CRC)	[172]

* in vivo experiments; FITC = fluorescein isothiocyanate; TS = tumour site; RC = rapid clearance; MS = median survival; TV = tumour volume; SAL = salinomycin; CHI = chitosan; HA = hyaluronic acid; HCPT = 10-hydroxycamptothecin; DATEG = di-amino-tri-ethylene glycol; Sh = short. AGSs = human gastric adenocarcinoma cell line; Ir = irinotecan; ⬆ = enhancing, enhanced, improving, improved, augmented, high, and higher; ⬇ = low, lower, decreased; reduced, and reduction; GC = gastric cancer; CRC = colorectal cancer.

**Table 10 ijms-26-09201-t010:** Modification/functionalization of CNTs through various molecules and their applications to treat gastrointestinal cancers.

CNTs	FM	Effectiveness	Tumour Model	Biocompatibility Test	Refs.
MWCNTs	PEG600, OP	Delayed cytotoxic activity	Colorectal cancer **	⬇ Toxicity	[160]
SWCNTs	PEG-10–10%PEI/Bcl-xL-shRNA, DOX	⬇ by 58-fold usual DOX IC_50_; ⬆ WS, ⬆ BFS, no PA	AGS GCCs/L929 **	⬇ Toxicity	[161]
SWCNTs	Chim/PEI/5-FU/CNT	⬆⬆⬆ TP, ⬆⬆⬆ DL, ⬇ invasion/proliferation, apoptosis	Gastric cancer **	⬆ Biocompatibility	[162]
SWCNTs	II-NCC	⬆⬆⬆ WD; ⬆ ACEs of CAP	Caco-2 colon **	⬆ Biocompatibility	[163]
SWCNTs	CpG	ACA in gliomas, ⬇ CCP, ⬇ invasion/migration	HCCsT116 **	⬆ Mice survival rate	[164]
SWCNTs	SN38, C225	Specific binding, controlled release of SN38, ⬆ ACEs	HCT116, HT29, SW620 **	⬆ Biocompatibility	[165]
MWCNTs	OP, PEG, SPIO	SR (56%, 144 h), ⬆ ACEs in vitro (96 h) ⬆ MRI signal	HCT116 **, BALB/c ANM *	⬇ Adverse effects	[152]
SWCNTs	HA, Ce6	⬆ ACEs, ⬆ WD, ⬆ early and late apoptosis	Caco-2 colon **	Possible ⬇ toxicity due to ⬆ WS	[171]

⬆ enhanced, improved, high, and higher; ⬇ low, lower, decreased, and reduced; HA = hyaluronic acid; Ce6 = chlorine e6; PEG = polyethylene glycol; WS = water solubility; DOX = doxorubicin; PA = particle aggregation; OP = oxaliplatin; FMs = functionalizing molecules; * in vivo experiments ** in vitro experiments; DL = drug loading; FA = folic acid; PEI = polyethylene imine; pDNA = plasmid DNA; Chim = chimera; 5-FU = 5-fluorouracil; II-NCC = N, carboxy cellulose II; CAP = capecitabine; ⬆⬆⬆ strongly enhanced, increased, and suppression; WS = water solubility; WD = water dispersion; BFS = biological fluids dispersity; CT = combination therapy; TP = tissue permeability; N.R. = not reported. CpG = complex; ACA = anticancer activity; HCCsT116 = human colon cancer cell line T116; CCP = cancer cell proliferation; ACEs = anticancer effects; SPIO = superparamagnetic iron oxide; MRI = magnetic resonance imaging; SR = sustained release; ANM = athymic nude mice inoculated with HCT116 cells; N.R. = not reported.

**Table 11 ijms-26-09201-t011:** Approaches for limiting toxicity of CNTs [5].

Strategy	Goal	Modifying Agents/Methods	Results
CNTs surface modification with biocompatible materials or other molecules	⬆ Dispersion in biological fluids Influenced CU, ⬆ Solubility ⬇ Toxicity	Proteins, surfactants	⬆ TT, ⬆ TB, ⬇ Toxicity
		FA	⬆ In vivo tumour targeting, ⬆ Therapeutic benefits ⬇ Toxicity
		PA hydrogels *, biomaterial, TiO_2_	100% survival of L929 mouse fibroblast
Coatings of CNTs	⬆ CNTs biocompatibility ⬇ Potential toxicity Prevent direct contact with BS ⬆ CNTs solubility	Curcumin lysine **	⬇ IL-6, IL-8, IL-1β, TNFα, N-FκB ⬆ Antioxidant enzyme catalase, ⬇ ROS generation Recovery of MM, ⬇ Cell death
CNTs encapsulation CNTs to entrap BAM	⬇ Direct cells exposure to CNTs Control of CNTs release ⬇ CNTs impact on tissues	PEG (entrapping agent) Oxaliplatin (entrapped agent)	PEGylation delayed oxaliplatin release rate ⬆ Drug’s anticancer effects on HT-29 cells
Tailor Ø size and L	⬇ Toxicity	N.A.	⬆ SSA, ⬆ TM, ⬇ Toxicity, ⬇ Harm to lysosomes ***
Optimized PP	Remove MI Remove RC	Chemical/electrochemical oxidation High chlorine partial pressure MA digestion Incandescent annealing	⬇ Lower harmful effects
Engineering controls Suitable PPE	⬇ Inhalation	Proper ventilation/respiratory protection	⬇ Respiratory toxicity
CA with AO	⬇ OS ⬇ Damage to cells	Quercetin	Prevention of the oxidative damage ⬇ Inflammatory effects, ⬇ Immuno-toxic effects

* encapsulation agent for CNTs-COOH; ** used to coat MWCNTs; N.A. = not applicable; N.R. = not reported; ⬇ indicates minor reduction, lower, decreased, or decrease; ⬆ indicates improved, increase, increased, or major; PPE = personal protective equipment; OS = oxidative stress; BAM = bioactive molecules; Ø = diameter; L = length; PP = purification processes; CA = co-administration; AOs = antioxidants; CU = cellular uptake; MIs = metal impurities; RC = residual catalysts; BSs = biological systems; PA = polyacrylamide; FA = folic acid; MA = microwave assisted; TT = tumour targeting; TB = therapeutic benefits; SSA = specific surface area; TM = transmembrane mobility; MM = mitochondrial membrane; *** large Ø MWCNTs.

**Table 12 ijms-26-09201-t012:** Clinically approved nanoparticle-based treatments for cancer therapy.

Name	NPIs	API	AI	TGs
**Lipid-Based NPs**				
Doxil/Caelyx	LIP, PEG	Doxorubicin	FDA (1995), EMA (1996)	OC, HIV-KS, MM
Onivyde	LIP, PEG	Irinotecan	FDA (2015)	**MPC**
DaunoXome	LIP	Daunorubicin	FDA (1996)	HIV-KS
Myocet	LIP	Doxorubicin	EMA (2000)	MBC
Marqibo	LIP	Vincristine	FDA (2012)	PCNALLK
Mepact	LIP	Mifamurtide	EMA (2009)	NMR-OS
Vyxeos	LIP	Cytarabine/daunorubicin (5/1 M)	FDA (2017), EMA (2018)	HR-AMLK
Lipusu	LIP	Paclitaxel	China (2006)	BC, LC, OC
DepoCyt	LIP	Cytarabine	FDA (1999)	NM
**Polymer-based NPs**				
Oncaspar	PEG-P	ASNase	FDA (1994)	ALLK
Genexol-PM	Micelle	Paclitaxel	South Korea (2007)	MBC, **PC**
Eligard	PLGA	Leuprolide acetate	FDA (2002)	PRC
Neulasta	PEG-P	G-CSF	FDA (2002), EMA (2002)	NMMs
Zinostatin Stimalamer	SMA	NCS	Japan (1994)	**PUHCC**
**Albumin-based NPs**				
Abraxane	Albumin	Paclitaxel	FDA (2005), EMA (2008)	LC, MBC, **MPC**
Pazenir	Albumin	Paclitaxel	EMA (2019)	MBC, **MPAC**, NSCs-LC
**Other NPs**				
NanoTherm	F_2_O_3_	Not applicable	EMA (2011)	Glioblastoma
Hensify (NBTXR3)	HfO_2_	Radiotherapeutic	EMA (2019)	STS

NPIs = nanoparticle ingredients; HfO_2_ = Hafnium oxide; PEG-P = PEGylated protein; PEG = polyethylene glycol; SMA = shape memory alloy; PLGA = poly (lactic-co-glycolic acid); LIP = liposome; TGs = target cancers; API = active pharmaceutical ingredient; AI = approval year; FDA = US Food and Drug Administration; EMA = European Medicines Agency; ASNase = asparaginase; SMA = styrene-malic acid; NCS = neocarzinostatin; G-CSF = granulocyte colony stimulating factor; LC = lung cancer; MBC = metastatic breast cancer; MPC = metastatic pancreatic cancers; PUHCC = primary unresectable hepatocellular carcinoma; OC = ovarian cancer; OS = osteosarcoma; HIV-KS = HIV-associated Kaposi’s sarcoma; MM = multiple myeloma; PCNALLK = Philadelphia chromosome-negative acute lymphoblastic leukemia; NMR-OS = non-metastatic, resettable osteosarcoma; HR-AMLK = high-risk acute myeloid leukemia; ALLK = acute lymphoblastic leukemia; PRC = prostate cancer; NMM = non-myeloid malignancies; STS = soft-tissue sarcoma; MPAC = metastatic pancreatic adenocarcinoma; NSCs-LC = non-small-cell lung cancer; NM = neoplastic meningitis.

## Data Availability

No new data were created here.

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
