# Peer review of "From Early Diagnoses to New Treatments for Liver, Pancreatic, Gastric, and Colorectal Cancers Using Carbon Nanotubes: New Chances Still Underexplored"

_ijms, 2025, doi:10.3390/ijms26189201_

Round 1

Reviewer 1 Report (New Reviewer)

Comments and Suggestions for Authors

Alfei et al present a well-written and comprehensive review of carbon nanotubes for cancer diagnosis and therapy. The text is clearly written and supported by recent references. Nevertheless, there are some concerns I would like to address.

Major comments

  1. Abstract

The abstract is concise and aligned with the manuscript, but it would benefit from greater emphasis on novelty and specificity.

  1. Section 2 – Nanotechnology for improved cancer treatment and diagnosis

This section effectively introduces recent innovations in nanomaterials, but it reads broadly. To strengthen it, the discussion should be more focused on CNT-specific advantages, particularly their functionalization for targeted drug delivery in difficult-to-treat cancers such as triple-negative breast cancer. Incorporating clinical studies or ongoing trials—such as CNT-based biosensors for early detection or photothermal therapies—would enhance the translational perspective.

  1. Section 3 – Carbon nanotubes for anticancer therapy and diagnosis

This section highlights the multifunctional nature of CNTs in both therapeutic and diagnostic contexts. To deepen the analysis, a comparative table contrasting CNT toxicity with other nanomaterials would be valuable. Additionally, referencing outcomes from recent clinical studies would situate CNTs within the broader translational landscape. Expanding the discussion of limitations (toxicity, clinical lag, reproducibility) and providing benchmarks relative to alternative nanoplatforms would further strengthen the section.

  1. Section 4 – Hepatopancreatic cancer therapy and diagnosis using CNTs

This section is detailed, particularly with well-organized tables and examples. However, it remains largely preclinical. It would benefit from explaining why CNTs have not progressed to oncology clinics, unlike liposomes or polymeric nanoparticles. A comparative discussion of translational barriers across nanomaterials would be instructive. Toxicity should be more critically analyzed, ideally through a table comparing CNT profiles with other nanoparticles. Case studies of CNT-based therapies could also be enriched by comparing them with viral vector gene delivery or other established systems.

  1. Section 5 – Carbon nanotubes for gastric and colorectal cancer therapy and diagnosis

The section is clear and supported by specific case studies and tables. To ensure balance, however, it should include more discussion of preclinical or clinical toxicity data related to CNT use in these cancers.

  1. Section 6 – Main factors which hamper CNTs translation in clinical practice

This section identifies key translational barriers but remains somewhat descriptive. A more critical analysis would strengthen the section, especially if barriers specific to CNTs were compared with those of clinically advanced nanomaterials such as liposomes, polymeric nanoparticles, or gold nanoparticles. Without this comparison, it is difficult for readers to discern whether CNT challenges are unique or shared across the field of nanomedicine.

Minor comments

  • Ensure consistency in terminology throughout (eg, always use “carbon nanotubes [CNTs]” on first mention, then CNTs).
  • Strengthen cross-referencing across sections—for example, linking toxicity discussions in Sections 3–5 more explicitly to the translational barriers presented in Section 6.

Author Response

Alfei et al present a well-written and comprehensive review of carbon nanotubes for cancer diagnosis and therapy. The text is clearly written and supported by recent references. Nevertheless, there are some concerns I would like to address.

We thank the Reviewer for these positive comments. We hope to satisfy all other requests from this Reviewer, to further enhance the quality of our paper.

Major comments

  1. Abstract

The abstract is concise and aligned with the manuscript, but it would benefit from greater emphasis on novelty and specificity.

We thank the Reviewer for this suggestion, which has allowed us to better evidence the novelty and specificity of this study. The abstract has been substantially modified. Please, see lines 11-43.

  1. Section 2 – Nanotechnology for improved cancer treatment and diagnosis

This section effectively introduces recent innovations in nanomaterials, but it reads broadly. To strengthen it, the discussion should be more focused on CNT-specific advantages, particularly their functionalization for targeted drug delivery in difficult-to-treat cancers such as triple-negative breast cancer. Incorporating clinical studies or ongoing trials—such as CNT-based biosensors for early detection or photothermal therapies—would enhance the translational perspective.

We thank the Reviewer for these suggestions. In this regard, a discussion on the possible use of CNTs to treat triple-negative breast cancer (TNBC) and on the application of functionalized CNTs to realize the targeted drug delivery in TNBC has been added in Section 2, including two new references. Please, see lines 181-193. Concerning possible clinical studies or ongoing trials regarding the use of CNTs to detect or treat cancers, they were already reported and discussed in the original paper in the dedicated Section 6.2 (main text) (lines 164-191, Section 6.2.) associated to Tables S2 and S3 in the Supplementary Materials file.  

  1. Section 3 – Carbon nanotubes for anticancer therapy and diagnosis

This section highlights the multifunctional nature of CNTs in both therapeutic and diagnostic contexts. To deepen the analysis, a comparative table contrasting CNT toxicity with other nanomaterials would be valuable. Additionally, referencing outcomes from recent clinical studies would situate CNTs within the broader translational landscape. Expanding the discussion of limitations (toxicity, clinical lag, reproducibility) and providing benchmarks relative to alternative nanoplatforms would further strengthen the section.

We thank the Reviewer for these suggestions. In this regard, a new Table has been created and included in the main text as Table 1 with a brief discussion. As asked, it compares the toxic effects observed in several in vitro and in vivo studies carried out on different cell lines and animal models of three different CNTs-based materials (rows 8-10), with those of fullerene and several inorganic nanoparticles commonly applied in nanomedicine (Table 1, lines 316-341). Regarding outcomes from recent clinical trials, as reported in our response to the previous point, they were already present in the original paper in the dedicated Section 6.2 in the main text (lines 164-191, Section 6.2.) and in Tables S2 and S3 in the Supplementary Materials file. A discussion of limitations (toxicity, clinical lag, reproducibility) was already present in the not revised paper (Section 6). Anyway, to follow the suggestion of the Reviewer, it was expanded, and a new Figure (Figure 6, revised version) was included. Please, see lines 34-136, Section 6. Additionally, Section 6.1. was created were the tactics developed to reduce CNTs toxicity were reported and discussed. Please, see lines 137-157 and Table 11. Anticipations on the existence of these sections has been now inserted in Section 3 in lines 330-332.

  1. Section 4 – Hepatopancreatic cancer therapy and diagnosis using CNTs

This section is detailed, particularly with well-organized tables and examples. However, it remains largely preclinical. It would benefit from explaining why CNTs have not progressed to oncology clinics, unlike liposomes or polymeric nanoparticles. A comparative discussion of translational barriers across nanomaterials would be instructive.

We thank the Reviewer for these useful comments. The information reported in this paper concerning the experimentation of CNTs-based devices for detecting and treating hepatopancreatic tumours remains preclinical because, currently, only two clinical trials have been reported. Three other clinical trials regarded the management of other types of tumours, and additional two did not regard tumours. This information is included in Section 6.2. (lines 164-191). The explanation of why CNTs have not still progressed to oncology clinics, unlike liposomes or polymeric nanoparticles, is discussed in Section 6. Please, see lines 34-136, Section 6.

Toxicity should be more critically analyzed, ideally through a table comparing CNT profiles with other nanoparticles.

As reported in Point 3, we have included in the main text Table 1 with related discussion, which should satisfy this request from the Reviewer (lines 316-341).

Case studies of CNT-based therapies could also be enriched by comparing them with viral vector gene delivery or other established systems.

The required comparisons are already present in Table 4 (revised paper) (rows 8 and 11) and in lines 142-143, 162-163, 245-248, Figure 4, 187-191 (Section 5) and Table 10 (revised paper, row 2, Section 5). Anyway, to satisfy this Reviewer an additional part has been inserted in lines 79-99.

  1. Section 5 – Carbon nanotubes for gastric and colorectal cancer therapy and diagnosis

The section is clear and supported by specific case studies and tables. To ensure balance, however, it should include more discussion of preclinical or clinical toxicity data related to CNT use in these cancers.

Although already presented in Table 10, column 4, the required information is now also included in the main text (Section 5.3, lines 30-33).

  1. Section 6 – Main factors which hamper CNTs translation in clinical practice

This section identifies key translational barriers but remains somewhat descriptive. A more critical analysis would strengthen the section, especially if barriers specific to CNTs were compared with those of clinically advanced nanomaterials such as liposomes, polymeric nanoparticles, or gold nanoparticles. Without this comparison, it is difficult for readers to discern whether CNT challenges are unique or shared across the field of nanomedicine.

As pointed out previously, this Section has been expanded to better satisfy the Reviewer’s requests.

Minor comments

  • Ensure consistency in terminology throughout (eg, always use “carbon nanotubes [CNTs]” on first mention, then CNTs).

Done

  • Strengthen cross-referencing across sections—for example, linking toxicity discussions in Sections 3–5 more explicitly to the translational barriers presented in Section 6.

Done.

The English could be improved to more clearly express the research.

The manuscript was revised by Prof. Deirdre Kantz, English mother tongue teacher and our colleague, working for the University of Genoa and Pavia.

Reviewer 2 Report (New Reviewer)

Comments and Suggestions for Authors

The manuscript entitled: “From Early Diagnosis to New Treatments for Liver, Pancreatic, 2 Gastric and Colorectal Cancers Using Carbon Nanotubes: New 3 Chances Still Too Little Explored” by Silvana Alfei and coworkers has been revised. The manuscript is well-written and sounds interesting for the scientific community International Journal of Molecular Sciences. However, some aspects of the manuscript should be improved and I feel that the paper needs a major revision before it can be published in the journal.

  1. Table 3, Table 4, I think it is important to include a column in the table mentioning the functionalizing molecules, as they constitute the specificity of the therapy with these CNTs.
  2. 1.3; 4.4; 5.3 Authors' Considerations, I think these sections can be moved to section 7.
  3. Section 7. The authors mention a Statistical Analysis of Results and Authors' Considerations. I am of the opinion that the authors did not perform a statistical analysis. Creating graphs in Excel does not mean that a statistical analysis was performed. It could be called "Section 7. Authors' Considerations" and include sections 4.1.3; 4.4; 5.3 here.
  4. Section 7, lines 146-154 is the methodology adopted for the development of this review article; it could be placed in another section, e.g. at the end of the introduction.
  5. The paper has no section with a conclusion. While there are some ideas that could be considered the conclusion of the paper in section 7, I think a section devoted to the conclusions of this manuscript is important.
  6. Line 126 “market[30,35,36],.” Please check the entire manuscript, as a space is missing in the in-text references.
  7. Line 179 “sectors [66]..”
  8. Line 232: The abbreviations PTT and PDT have not been described.
  9. Line 376 “..”
  10. Line 73: The abbreviation had already been defined previously (carbon nanotubes (CNTs)).
  11. Line 217: “rat model [138]. SOR@f-CNTs microcapsules…” One point, as they are part of the same concept.
  12. Line 224: “complex [56]. When administered …”A period as they are part of the same concept.

Author Response

The manuscript entitled: “From Early Diagnosis to New Treatments for Liver, Pancreatic, Gastric and Colorectal Cancers Using Carbon Nanotubes: New Chances Still Too Little Explored” by Silvana Alfei and coworkers has been revised. The manuscript is well-written and sounds interesting for the scientific community International Journal of Molecular Sciences. However, some aspects of the manuscript should be improved, and I feel that the paper needs a major revision before it can be published in the journal.

We thank the Reviewer for this first positive comments. We hope to satisfy all other requests from this Reviewer to further enhance the quality of our paper.

  1. Table 3, Table 4, I think it is important to include a column in the table mentioning the functionalizing molecules, as they constitute the specificity of the therapy with these CNTs.

We thank the Reviewer for this suggestion, which has given us the opportunity to better specify which are the functionalizing molecules used to engineer the CNTs based antitumor agents reported in Table 3 and 4. In fact, in both Tables, the functionalizing molecules were already inserted in the names of constructed devices, reported in column 1. Anyway, since not evidenced, they could go unnoticed. Sorry for this inconvenience. Now, without adding a new column, which, in our opinion, is not necessary, we have underlined the abbreviations of the functionalizing molecules present in the names of anticancer devices (first column of each Table). Abbreviations for these molecules were already specified in the footnotes of both Tables. We hope that this solution could satisfy the Reviewer.

  1. 1.3; 4.4; 5.3 Authors' Considerations, I think these sections can be moved to section 7.

We thank the Reviewer for this suggestion, which has given us the opportunity to better clarify our choice of positioning sub-sections 4.1.3., 4.4. and 5.3 in the current positions. The discussion reported in such Sections are strictly correlated with data and case studies reported in the respective previous sections. Therefore, separating them from their context and grouping them all together in Section 7 could cause confusion for the reader. However, the Reviewer made this rational suggestion because the titles, we mistakenly gave to these sections, "Authors' Considerations", immediately recall the title of Section 7, which would seem appropriate to contain them. This is not the case, and we apologize for having caused doubt and confusion for the reader, here as Reviewer. We have therefore changed the titles of the three sections in “Discussion on the Case Studies Previously Reported”. We hope that this solution could satisfy the Reviewer.

  1. Section 7. The authors mention a Statistical Analysis of Results and Authors' Considerations. I am of the opinion that the authors did not perform a statistical analysis. Creating graphs in Excel does not mean that a statistical analysis was performed. It could be called "Section 7. Authors' Considerations" and include sections 4.1.3; 4.4; 5.3 here.

The Reviewer is right concerning her/his comment on the statistical valence of bar graphs in Section 7, and its title has been changed according to her/his suggestion. Anyway, for the reasons explained in the previous point, we ask kindly the Reviewer to not force us to move 4.1.3., 4.4., and 5.3. in Section 7. We have changed the titles not correctly assigned to these Sections. We hope that this solution could satisfy the Reviewer.

  1. Section 7, lines 146-154 is the methodology adopted for the development of this review article; it could be placed in another section, e.g. at the end of the introduction.

The suggestion from the Reviewer has been satisfied. Please see lines 117-134.

  1. The paper has no section with a conclusion. While there are some ideas that could be considered the conclusion of the paper in section 7, I think a section devoted to the conclusions of this manuscript is important.

The suggestion from the Reviewer has been satisfied. A Conclusion Section 8 have been added. Please see lines 292-336 in Section 8.

  1. Line 126 “market[30,35,36],.” Please check the entire manuscript, as a space is missing in the in-text references.

We apologise to the Reviewer for the inconvenience of spaces missing between words and citations. We have already experienced this issue. It is a problem which derives from using Mendeley as citing tool. As long as Mendeley mode is active, the software rejects the insertion of spaces. If you manually insert them, every time you reopen the file, the spaces will be removed again, even if you save the corrections. We assure the Reviewer that during the proofreading phase, if the paper will be accepted, the Editorial Office will resolve this issue, as made for other our articles.

  1. Line 179 “sectors [66]..”

Corrected. Since the paragraph has been moved, now old ref. 66 is ref. 53. Please, see line 167.

  1. Line 232: The abbreviations PTT and PDT have not been described.

Sorry. Now they have been described in the previous lines 195-196.

  1. Line 376 “..”

Corrected. Line 442.

  1. Line 73: The abbreviation had already been defined previously (carbon nanotubes (CNTs)).

Corrected. Lines 73-74.

  1. Line 217: “rat model [138]. SOR@f-CNTs microcapsules…” One point, as they are part of the same concept.

Corrected. Lines 235-241.

  1. Line 224: “complex [56]. When administered …”A period as they are part of the same concept.

Corrected. Lines 245-252.

English language: The English is fine and does not require any improvement.

We thank a lot the Reviewer for having appreciated our language style.

Reviewer 3 Report (New Reviewer)

Comments and Suggestions for Authors

This paper presents an interesting and comprehensive review on the use of carbon nanotubes (CNTs) for the detection and treatment of various types of cancer. It covers a broad range of applications and includes a significant portion of the relevant literature on this topic. The manuscript is generally well-structured and informative. However, there are several points that should be addressed by the authors to enhance the overall quality, clarity, and scientific rigor of the article.

1) The paper is too long and unfocused, which makes it difficult to maintain a clear understanding of the main issues being discussed. A more concise and structured approach would help improve clarity and reader engagement. 
2) The abstract lacks of crucial results and important details (such as information on nanostructure toxicity and key findings) and should be revised accordingly. I also recommend further improvement of the English language throughout the paper: for example, the phrase 'researchers have tried to 17 address using nanomaterials (NMs)' is awkward, and should be rephrased for better readability.
3) The Introduction is very dispersive and includes a considerable amount of redundant information. In my opinion, it should be streamlined to focus only on the most relevant and useful details
4) While the potentialities of this technology for successful cancer detection (particularly as sensors) is known and encouraging, the manuscript does not clearly explain how it could be effectively applied in safe cancer treatment, especially given the current concerns regarding its toxicity.
5) Moreover, the manuscript does not clearly present any effective solutions to address the toxicity of these technologies, which, in my opinion, represents one of the key challenges for their future clinical application.

Author Response

This paper presents an interesting and comprehensive review on the use of carbon nanotubes (CNTs) for the detection and treatment of various types of cancer. It covers a broad range of applications and includes a significant portion of the relevant literature on this topic. The manuscript is generally well-structured and informative. However, there are several points that should be addressed by the authors to enhance the overall quality, clarity, and scientific rigor of the article.

We are very thankful to the Reviewer for this positive comment. We will try to address her/his further comments to further ameliorate our paper.

1) The paper is too long and unfocused, which makes it difficult to maintain a clear understanding of the main issues being discussed. A more concise and structured approach would help improve clarity and reader engagement.

Thank for this comment. The topic of this paper was very vast. Although literature on the use of CNTs to treat and diagnose the types of tumours here considered (PC, LC, GC and CRC) is limited compared to that concerning the most part of known cancers, several case studies exist, mainly if both cancer therapy and cancer diagnosis is considered in the same paper. So that, it was impossible for us to reduce and concentrate too much the related information. The risk could be that of omitting essential information. The work is long, we recognize that, but in our opinion is well structured, especially after revisions required by other 3 Reviewers. Specifically, we have structured our review as a series of concentric information containers, that go from the outermost to the innermost, in which the information reported becomes from more general to more specific, as one move into the successive innermost containers. So, after an Introduction providing readers a general background to understand the current scenario concerning tumours here considered (PC, LC, GC and CRC), the current existing approaches for their detection and treatments and main associated issues, as well as a brief part on methods followed to collect material for this study, a sequence of other Sections follows, whose information becomes more and more specific. First, there is an introductive part on the use of nanotechnology in general for improved cancer treatment and diagnosis, ending on a discussion of the advantages and disadvantages to use CNTs rather than other nanomaterials. Then, the specific part on CNTs starts, including the use of CNTs to detect and treat hepatopancreatic cancers, which are tumours affecting organs strictly correlated. This part was divided into rational subsections, where several case studies were reported using Tables, which were discussed. A structure like this was used also when the use of CNTs to detect gastric cancer and colorectal cancer was considered, which are tumours affecting organs strictly correlated, again. Section 6 follows on the challenging obstacles, which still limit CNTs translation in clinical practice, tactics developed to limit their possible toxicity and CNTs research, which is closer to translate in clinical setting, reporting clinical trials already developed. Sections 7 and 8 follow, which contain a numerical and critical analyses of reported results and the authors conclusions and recommendations, respectively.

In our opinion, as previously commented by the same Reviewer, this review covers a broad range of applications and includes a significant portion of the relevant literature on this topic, thus resulting well-structured and informative.

2) The abstract lacks of crucial results and important details (such as information on nanostructure toxicity and key findings) and should be revised accordingly. I also recommend further improvement of the English language throughout the paper: for example, the phrase 'researchers have tried to address using nanomaterials (NMs)' is awkward, and should be rephrased for better readability.

Crucial results and important details concerning the use of CNTs to treat tumours and specifically PC, LC, GC and CRCs, as well as advantages and disadvantages deriving by their applications can only generally be introduced in the abstract, and in these terms, they have been evidenced. A general information on the scenario concerning the CNTs toxicity has instead been evidenced in lines 39-40. Collectively, the abstract has been substantially modified. Please, see lines 11-43.

The sentence signalled by the Reviewer has been reformulated (lines 18-19). Generally, the manuscript was revised by Prof. Deirdre Kantz, English mother tongue teacher and our colleague, working for the University of Genoa and Pavia.

3) The Introduction is very dispersive and includes a considerable amount of redundant information. In my opinion, it should be streamlined to focus only on the most relevant and useful details

We regret to contrast this Reviewer opinion, which in turn contrasts the first comment of the same Reviewer (The manuscript is generally well-structured and informative). Anyway, since the Introduction accounts for only 68 lines, in our opinion, it cannot be dispersive and contain a considerable amount of redundant information. Conversely, it provides readers a general background to understand the current scenario concerning tumours here considered (PC, LC, GC and CRC), the current existing approaches for their detection and treatments and the main associated issues. We kindly ask the Reviewer to not force us to reduce it, with the risk of omitting essential introductive information for the readers.

4) While the potentialities of this technology for successful cancer detection (particularly as sensors) is known and encouraging, the manuscript does not clearly explain how it could be effectively applied in safe cancer treatment, especially given the current concerns regarding its toxicity.

Concerning the request of Reviewer, Figure 1, is an initial schematic exemplification of how CNTs could be applied both in efficient sensing and treating cancer, applications that, as explained in the paper, are strictly connected one to each other. Moreover, how CNTs could be applied for safe cancer treatment is described in several points on this review. Among others, the Reviewer can consider the parts evidenced in yellow in lines 252-273, 280-291, 303-312 and Table 4 in the revised paper (column 2). In Section 4.2., the Reviewer should consider the new lines 79-97 and the already existing parts in lines 112-114, 118-122, 133-136 and 142-143. In Section 4.3., lines 163-163, 167-169, 172-176, 180-183, 187-188, Table 5 (revised paper, column 2), lines 201-206, 213-215, 235-238, 245-248 and Table 7 (revised paper, column 5) should be considered. In Section 4.4., lines 5-9 contain the desired information, while in Section 5.2, lines 134-137, 140-148, 154-162, 169-173, 179-182, 187-191, 197-201, 209-217, etc, and Table 9 (column 2), as well as Table 10 (revised paper, columns 2 and 5) could be considered.

5) Moreover, the manuscript does not clearly present any effective solutions to address the toxicity of these technologies, which, in my opinion, represents one of the key challenges for their future clinical application.

The information required by the Reviewer is reported as an anticipation, in lines 203-209 and lines 17-19 (Section 4.4.). Anyway, a part concerning effective tactics to address nanotoxicity of CNTs have been created. Please, consider lines 137-157 and new Table 11, in Section 6.1 (revised manuscript).

The English could be improved to more clearly express the research.

The English language has been improved as previously described.

Reviewer 4 Report (New Reviewer)

Comments and Suggestions for Authors 1. Figures 1 and 2 should be properly referenced, and since they are not the authors’ own, they should be labeled with Roman numerals. 2.It would strengthen the manuscript if a comparison of the efficacy of different cancer diagnostic approaches using carbon nanotubes were included. 3.The basis on which the review data were collected is not stated. This should be clarified and placed immediately after the general introduction. 4.The manuscript lacks a Recommendation section, which should be added to highlight future perspectives.

Author Response

  1. Reviewer 4

    1. Figures 1 and 2 should be properly referenced, and since they are not the authors’ own, they should be labeled with Roman numerals.

    We make kindly note to the Reviewer that Figure 1 and 2 are already referenced, both in the text and in the Figure captions. The reference is Ref. 1. Concerning labels, we used the Arabic numerals according to IJMS instructions.

    2.It would strengthen the manuscript if a comparison of the efficacy of different cancer diagnostic approaches using carbon nanotubes were included.

    We make kindly note to the Reviewer, that comparisons on the efficacy of different carbon nanotubes in terms of detection range, linear range, LOD and detection times are already present Sections 4.1.1., 4.1.2., Section 5.1 and in Tables 2, 3 and 8 (revised paper).

    3.The basis on which the review data were collected is not stated. This should be clarified and placed immediately after the general introduction.

    The suggestion of the Reviewer has been satisfied inserting the new subsection 1.1. at the end of Introduction. Please see lines 117-134.

    4.The manuscript lacks a Recommendation section, which should be added to highlight future perspectives.

    The section has been included as Section 8. Conclusions and Recommendations. Please see lines 292-336 in Section 8.

    The English is fine and does not require any improvement.

    We thank the Reviewer for this positive comment on our English.

Round 2

Reviewer 1 Report (New Reviewer)

Comments and Suggestions for Authors The authors have thoroughly addressed all previously raised concerns and significantly enhanced the manuscript's quality. The revised work is suitable for consideration for publication in this journal.

Reviewer 2 Report (New Reviewer)

Comments and Suggestions for Authors

Dear author, thank you for taking my comments into consideration. I believe the appropriate changes have been made. I have no comments on your article.
Best regards.

Reviewer 3 Report (New Reviewer)

Comments and Suggestions for Authors

In my opinion, this second version of the manuscript titled "From Early Diagnosis to New Treatments for Liver, Pancreatic, Gastric, and Colorectal Cancers Using Carbon Nanotubes: New Chances Still Too Little Explored" has been significantly improved. The authors have clarified several key aspects and included important additional details as previously requested. As a result, I find the current version of the manuscript suitable for publication in this journal.

Reviewer 4 Report (New Reviewer)

Comments and Suggestions for Authors

Thanks for doing modifications 

This manuscript is a resubmission of an earlier submission. The following is a list of the peer review reports and author responses from that submission.

Round 1

Reviewer 1 Report

Comments and Suggestions for Authors

Dear Authors,

The manuscript is an interesting piece of work; however, I have the following comments:

  1. Embedding hyperlinks directly in the manuscript suggests a copy‑and‑paste approach and indicates insufficient attention to document preparation.
  2. Several sections of the text lack any bibliographic citations.
  3. Some references are missing (for example, reference 23 does not appear), while others (e.g., reference 31) are cited before they should be.
  4. The quality of the figures is poor. Please replace or enhance them to ensure clarity.
  5. The use of the “@” symbol is inappropriate; instead, each molecule should be described with a clear, structured, and precise text label.
  6. While I appreciate the approach in Section 5, Section 6 is inadequately drafted. Referring only in passing to supplementary material diminishes its importance—this could be a key section if developed in greater detail.
  7. Avoid first‑person or colloquial expressions such as “Anyway, we want to sensitize researchers to pay more attention to…”

I hope these comments help improve the clarity and professionalism of your manuscript.

Author Response

Dear Authors,

The manuscript is an interesting piece of work; however, I have the following comments:

  1. Embedding hyperlinks directly in the manuscript suggests a copy‑and‑paste approach and indicates insufficient attention to document preparation.

We thank the Reviewer for her/his suggestion. Sorry, for our carelessness. All hyperlinks have been removed from the main text and replaced with references numbers in the style of IJMS. Please, you can find these changes in the captions of Figure 1-5 (Refs 64 and 96) and in lines 103-107 of Section 6.1 (revised version).

  1. Several sections of the text lack any bibliographic citations.

We have carefully checked all the manuscript, and the due additional references have been inserted, which are evidenced in yellow. Please, consider Ref 23, 24, 29, 30, 32, 33, 33 , 35 and the already existing Ref. 5 which has been added in lines 220, 222.

  1. Some references are missing (for example, reference 23 does not appear), while others (e.g., reference 31) are cited before they should be.

We thank the Reviewer for this comment. Sorry, for these mistakes, which have been solved. References have been updated. Now reference list corresponds to references in the text and references in the text appear in the correct numerical order.

  1. The quality of the figures is poor. Please replace or enhance them to ensure clarity.

Since the resolution of Figures 1-6 already respected the IJMS requirements, to address the Reviewer request, their quality has been further improved using the instruments provided by Microsoft 365 to correct and ameliorate pictures.

  1. The use of the “@” symbol is inappropriate; instead, each molecule should be described with a clear, structured, and precise text label.

We thank the Reviewer for this comment. Anyway, we make kindly note her/him that such annotation is not our invention. We have used the “@” notation to connect the main components (ingredients) of nanotubes composites, because it is a notation extensively adopted by authors who publish in this sector. We kindly ask the Reviewer to accept this type of notation, since scientifically accepted.

Please, consider:

10.1016/J.ACA.2017.12.030,  10.1016/J.IJBIOMAC.2019.09.176, https://doi.org/10.1016/j.apsusc.2024.160405, https://doi.org/10.1016/j.surfin.2020.100672, https://doi.org/10.1016/j.ijhydene.2016.09.085, etc.

  1. While I appreciate the approach in Section 5, Section 6 is inadequately drafted. Referring only in passing to supplementary material diminishes its importance—this could be a key section if developed in greater detail.

We agree with the Reviewer and thank her/him for this suggestion. In this regard, the material related to the old Section 6 (Section 7 in the revised paper), previously relegated to Supplementary Materials has been moved in the main text of the new Section 7, to give it more importance. All related parts have been modified and updated. Please, consider the new structure of Section 7 and deleted lines 195-211 (Supplementary Materials descriptions).

  1. Avoid first‑person or colloquial expressions such as “Anyway, we want to sensitize researchers to pay more attention to…”

First- person and colloquial expressions have been removed by all manuscript. We thank the Reviewer for this suggestion. Some examples can be found at lines 28, 31, 106, 107 185-186, 192-194, 197, 199, 202, line 30 page 23, line 275, page 29, etc.

Reviewer 2 Report

Comments and Suggestions for Authors

In this review, the authors have indicated that pancreatic, liver, gastric, colorectal, and rectal cancers are highly lethal, with low 5-year survival rates due to late diagnosis, limited treatment options, recurrence, and drug resistance. CNTs have shown great promise in cancer diagnosis and therapy, acting as effective vehicles for targeted treatments while improving therapeutic outcomes and minimizing toxic effects. Despite their potential, CNT-based applications for these cancers remain underexplored. Overall, this review highlights the current advancements, emphasizes the need for further research, and addresses challenges such as toxicity, regulatory barriers, and clinical translation to improve early diagnosis and treatment of these aggressive cancers. This review is recommended for the publication in IJMS after revision. Below are several comments for the revision.

  1. The section titled “1.1. Nanotechnology for Improved Cancer Treatment and Diagnosis” can be revised to “2. Nanotechnology for Improved Cancer Treatment and Diagnosis”.
  2. For the section “5.1. CNTs Technologies Closest to the Clinical Application”, the limitations of CNTs for clinical use should be described. What are the reasons CNTs have not been approved by the FDA?
  3. In the section 1.1, the authors have indicated “Eco-friendly and biocompatible nanocomposites have demonstrated improved therapeutic efficacy…”, additional literatures can be added to emphasize their importance.

https://doi.org/10.1186/s12951-023-02208-3

Author Response

In this review, the authors have indicated that pancreatic, liver, gastric, colorectal, and rectal cancers are highly lethal, with low 5-year survival rates due to late diagnosis, limited treatment options, recurrence, and drug resistance. CNTs have shown great promise in cancer diagnosis and therapy, acting as effective vehicles for targeted treatments while improving therapeutic outcomes and minimizing toxic effects. Despite their potential, CNT-based applications for these cancers remain underexplored. Overall, this review highlights the current advancements, emphasizes the need for further research, and addresses challenges such as toxicity, regulatory barriers, and clinical translation to improve early diagnosis and treatment of these aggressive cancers. This review is recommended for the publication in IJMS after revision. Below are several comments for the revision.

We thank the Reviewer for having appreciated our paper.

  1. The section titled “1.1. Nanotechnology for Improved Cancer Treatment and Diagnosis” can be revised to “2. Nanotechnology for Improved Cancer Treatment and Diagnosis”.

We thank a lot the Reviewer for this suggestion which has been addressed. Please, consider line 111.

  1. For the section “5.1. CNTs Technologies Closest to the Clinical Application”, the limitations of CNTs for clinical use should be described. What are the reasons CNTs have not been approved by the FDA?

The reasons for which CNTs have not approved by the FDA have been included in the new Section 6.1. at lines 78-81.

.

  1. In the section 1.1, the authors have indicated “Eco-friendly and biocompatible nanocomposites have demonstrated improved therapeutic efficacy…”, additional literatures can be added to emphasize their importance.

https://doi.org/10.1186/s12951-023-02208-3

Additional references included that suggested by the Reviewer have been inserted in the test and list of references.